# Transformers learn through gradual rank increase

**Enric Boix-Adserà**[*1,2]    **Etai Littwin**[*1]
**Emmanuel Abbe**[1,3]    **Samy Bengio**[1]    **Joshua Susskind**[1]
[1]Apple    [2]MIT    [3]EPFL
eboix@mit.edu,emmanuel.abbe@epfl.ch
{elittwin,bengio,jsusskind}@apple.com

## Abstract

We identify incremental learning dynamics in transformers, where the difference between trained and initial weights progressively increases in rank. We rigorously prove this occurs under the simplifying assumptions of diagonal weight matrices and small initialization. Our experiments support the theory and also show that phenomenon can occur in practice without the simplifying assumptions.

## 1   Introduction

The transformer architecture achieves state of the art performance in various domains, yet we still lack a solid theoretical understanding of its training dynamics [VSP+17, DCLT19, LOG+19, DBK+20]. Nevertheless, the theoretical toolbox has matured over the last years and there are promising new approaches. One important line of work examines the role that initialization scale plays on the trajectory taken by gradient descent [JGH18, COB18, GSJW19, MGW+20, JGS+21, SS21, KC22]. When the weights are initialized small, it has been shown for simple networks that an *incremental learning* behaviour occurs, where functions of increasing complexity are learned in stages. This regime is known to be richer than the large-initialization regime[1], but the incremental learning dynamics are difficult to analyze, and are so far understood only for extremely simple architectures. Can we apply this analysis to transformers? Namely:

*Are there incremental learning dynamics when training a transformer architecture?*

An obstacle is that past work on incremental learning has mainly studied linear networks [Ber22, ACHL19, MKAA21, LLL20, WGL+19, JGS+21, GSSD19, SKZ+23, PF23], with one paper studying nonlinear 2-layer fully-connected networks [BPVF22]. In contrast, transformers have nonlinear attention heads that do not fall under previous analyses: given $\boldsymbol{X} \in \mathbb{R}^{n \times d}$, an attention head computes

$$\mathrm{attention}(\boldsymbol{X}; \boldsymbol{W}_K, \boldsymbol{W}_Q, \boldsymbol{W}_V, \boldsymbol{W}_O) = \mathrm{smax}(\boldsymbol{X}\boldsymbol{W}_K\boldsymbol{W}_Q^\top\boldsymbol{X}^\top)\boldsymbol{X}\boldsymbol{W}_V\boldsymbol{W}_O^\top \qquad (1)$$

where $\boldsymbol{W}_K, \boldsymbol{W}_Q, \boldsymbol{W}_V, \boldsymbol{W}_O \in \mathbb{R}^{d \times d'}$ are trainable matrices, and the softmax is applied row-wise. A transformer is even more complex, since it is formed by stacking alternating layers of attention heads and feedforward networks, along with residual connections.

**Main finding**   Our main finding is that transformers exhibit incremental learning dynamics, where *the difference between the trained and initial weights incrementally increases in rank*. Our results have a theoretical component and an experimental component.

---

[1]In the large-initialization regime, deep learning behaves as a kernel method [JGH18, COB18]. Various separations with kernels are known for smaller initialization: e.g., [GMMM19, ABM22, MKAS21].

37th Conference on Neural Information Processing Systems (NeurIPS 2023).

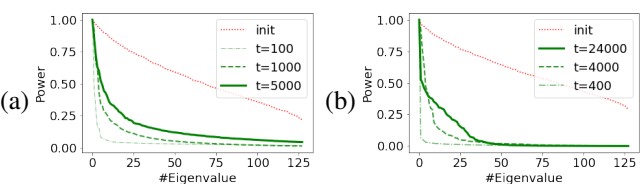

Figure 1: For an attention head in ViT trained on (a) CIFAR-10, and (b) ImageNet, we plot the normalized spectra of $W_K W_Q^\top$ at initialization (in red), and of the learned perturbations to $W_K W_Q^\top$ at different iterations (in green).

**Theoretical contributions**   For our theory, we study a simplification of the transformer architecture, where the attention head weights are diagonal matrices: i.e., in each attention head we have $W_K = \mathrm{diag}(w_K)$, where $w_K \in \mathbb{R}^d$ are trainable weights, and similarly for $W_Q, W_V$ and $W_O$. We rigorously establish the training dynamics of this architecture under gradient flow when the initialization is small. We prove that dynamics occur in discrete stages: (1) during most of each stage, the loss plateaus because the weights remain close to a saddle point, and (2) at the end, the saddle point is quickly escaped and the rank of the weights increases by at most one.

This theoretical result on transformers follows from a general theorem characterizing the learning dynamics of networks $f_{\mathsf{NN}}$ that depend on the product of parameters $u, v \in \mathbb{R}^p$ as

$$f_{\mathsf{NN}}(x; u, v) = h(x; u \odot v), \tag{2}$$

where $x$ is the input, $\odot$ denotes the elementwise product, and $h$ is a smooth function.

**Theorem 1.1** (Informal statement of incremental learning dynamics). *Let $f_{\mathsf{NN}}$ be a network of the form* (2)*, and suppose that the weights are initialized very small: i.e., the entries of $u, v$ are initialized on the order $\Theta(\alpha)$ for some small $\alpha > 0$. Then the dynamics of gradient flow training effectively proceeds in discrete stages, each one lasting time $\Theta(\log(1/\alpha))$. In each stage, the number of nonnegligible entries of $u \odot v$ increases by at most one.*

A transformer with diagonal weight matrices falls under this result when we only train the attention head weights. For example, if the transformer has one attention head, then we can take $u = [w_K, w_V] \in \mathbb{R}^{2d}$ and $v = [w_Q, w_O] \in \mathbb{R}^{2d}$ to be concatenations of the diagonal entries of the weights of the head; see Example 3.2 for more details and the extension to transformers with many heads. Then, using Theorem 1.1, we see that in each stage either $W_K W_Q^\top = \mathrm{diag}(w_K)\mathrm{diag}(w_Q)$ or $W_V W_O^\top = \mathrm{diag}(w_V)\mathrm{diag}(w_O)$ increases in effective rank by at most one.[2]

**Experimental contributions**   In our experiments, we first validate our theoretical results, which require the simplifying assumptions of small initialization and diagonal weight matrices.

Then, we conduct experiments on vision and language transformers in settings closer to practice, without any of the assumptions required by our theoretical analysis. Perhaps surprisingly, we again observe incremental learning dynamics, even though the assumptions of the theory are not met. The difference between trained and initial weights has low rank, and the rank of this difference grows gradually during training; see Figure 1. The incremental nature of the dynamics is easier to see for ImageNet, since for CIFAR-10 the rank of the weight difference does not grow as much.

## 1.1   Related work

**Relation to LoRA**   We note an intriguing connection to the LoRA algorithm, where a pretrained base model is cheaply fine-tuned by training a low-rank perturbation of the weights [LFLY18, AZG20, HSW+21]. The method is surprisingly powerful, and recently LoRA has been fundamental to allowing the open-source community to inexpensively fine-tune language models [PA23, TGZ+23]. On the other hand, in our work we observe that the trained weights are a low-rank perturbation of the initial weights due to the training dynamics, without having to apply an explicit rank constraint as in LoRA. This raises an exciting open question for future work: *can we explain and improve algorithms like LoRA by better understanding and quantifying the incremental dynamics of large transformers?*

---

[2]We also remark that Theorem 1.1 is interesting in its own right and may have other applications beyond transformers. It qualitatively recovers the incremental dynamics result of [Ber22, PF23] when specialized to linear diagonal networks, i.e., when $f_{\mathsf{NN}}(x; u, v) = \sum_{i=1}^{p} u_i v_i x_i$.

**Low-rank bias in nonlinear networks**  For 2-layer networks, it is known that low-rank bias in the weights emerges if the target function depends on a low-dimensional subspace of the input [ABM22, ABM23, DLS22, BBSS22, MHPG$^+$22]. The results of [ABM22, ABM23] are especially relevant, since they show that the rank of the weights increases in a sequential manner, determined by the "leap complexity" of the target function, which is reminiscent of our empirical observations on transformers. See also [FVB$^+$22, TVS23] for more investigations of low-rank bias in 2-layer networks under different assumptions. For transformers, [YW23] report that empirically the trained weights (using default initialization) are not low-rank. This is consistent with our claim that the difference between initial and trained weights is low-rank, since the initial weights might not be low-rank.

**Incremental learning dynamics**  Several works prove incremental learning behaviour in deep *linear* networks when the initialization is small. [GBLJ19] has shown that gradient descent dynamics on a 2-layer linear network with $L_2$ loss effectively solve a reduced-rank regression problem with gradually increasing rank. [GSSD19] prove a dynamical depth separation result, allowing for milder assumptions on initialization scale. [ACHL19, MKAA21] show implicit bias towards low rank in deep matrix and tensor factorization. [LLL20] show deep matrix factorization dynamics with small initialization are equivalent to a greedy low-rank learning (GLRL) algorithm. And [JGS$^+$21] independently provides a similar description of the dynamics, but without requiring balanced initialization. Finally, [Ber22, JLL$^+$23, PF23] overcome a technical hurdle from previous analyses by proving incremental learning for the entire training trajectory, rather than just the first stage. In contrast to our result, these prior works apply only to *linear* networks with certain convex losses, whereas our result applies to *nonlinear* networks. In order to make our extension to nonlinear networks possible, we must make stronger assumptions on the training trajectory, which we verify hold empirically. As far as we are aware, one other work on incremental learning handles nonlinear networks: [BPVF22] proves that a 2-layer network learns with a two-stage incremental dynamic; but that result needs the stylized assumption that all data points are orthogonal.

## 1.2  Paper organization

Sections 2, 3, and 4 contain theoretical preliminaries, definitions of the models to which our theory applies, and our main theoretical result on incremental dynamics. Section 5 provides experiments which verify and extend the theory. Section 6 discusses limitations and future directions.

## 2  Preliminaries

We consider training a network $f_{\mathsf{NN}}(\cdot; \boldsymbol{\theta})$ parametrized by a vector of weights $\boldsymbol{\theta}$, to minimize a loss

$$\mathcal{L}(\boldsymbol{\theta}) = \mathbb{E}_{\boldsymbol{x}, \boldsymbol{y}}[\ell(\boldsymbol{y}, f_{\mathsf{NN}}(\boldsymbol{x}; \boldsymbol{\theta}))],$$

where the expectation is over samples $(\boldsymbol{x}, \boldsymbol{y}) \in \mathbb{R}^{d_x} \times \mathbb{R}^{d_y}$ from a training data distribution, and $\ell : \mathbb{R}^{d_y} \times \mathbb{R}^{d_{out}} \to \mathbb{R}$. Consider a solution $\boldsymbol{\theta}(t)$ to the gradient flow[3]

$$\boldsymbol{\theta}(0) = \alpha \boldsymbol{\theta}_0, \quad \frac{d\boldsymbol{\theta}}{dt} = -\nabla_{\boldsymbol{\theta}} \mathcal{L}(\boldsymbol{\theta}) \tag{3}$$

where $\alpha > 0$ is a parameter governing the initialization scale, that we will take small. For our theory, we henceforth require the following mild regularity assumption on the loss and data.

**Assumption 2.1** (Regularity of data distribution and loss). The function $\ell(\boldsymbol{y}, \boldsymbol{\zeta})$ is continuously twice-differentiable in the arguments $[\boldsymbol{y}, \boldsymbol{\zeta}] \in \mathbb{R}^{d_y + d_{out}}$. There exists $C > 0$ such that almost surely the data is bounded by $\|\boldsymbol{x}\|, \|\boldsymbol{y}\| \leq C$.

The assumption on $\ell$ is satisfied in typical cases such as the square and the cross-entropy losses. The data boundedness is often satisfied in practice (e.g., if the data is normalized).

We also use the notation $\operatorname{supp}(\boldsymbol{a}) := \{i : a_i \neq 0\}$ to denote the support of a vector $\boldsymbol{a}$.

---

[3]Gradient flow training can be obtained as a limit of SGD or GD training as the learning rate tends to 0 (see, e.g., [Bac20]). It is a popular testbed for studying learning dynamics (see e.g., [SMG13, ACH18, RC20]), since is generally simpler to analyze than SGD.

# 3  Neural networks with diagonal weights

Our theory analyzes the training dynamics of networks that depend on products of diagonal weight matrices. We use $\odot$ to denote elementwise vector product.

**Definition 3.1.** A network $f_{\mathsf{NN}}$ is smooth with diagonal weights $\boldsymbol{\theta} = (\boldsymbol{u}, \boldsymbol{v}) \in \mathbb{R}^{2p}$ if it is of the form

$$f_{\mathsf{NN}}(\boldsymbol{x}; \boldsymbol{\theta}) = h(\boldsymbol{x}; \boldsymbol{u} \odot \boldsymbol{v})$$

where $h : \mathbb{R}^{d_x} \times \mathbb{R}^p \to \mathbb{R}^{d_{out}}$ is continuously twice-differentiable in its arguments in $\mathbb{R}^{d_x+p}$.

The assumption on $h$ precludes the use of the ReLU function since it is not continuously-differentiable. Otherwise the assumption is fairly mild since any $h$ can be used to express an architecture of any depth as long as the nonlinearities are twice-differentiable, which includes for example GeLUs (as used in ViT). We describe how to express a transformer with diagonal weights.

**Example 3.2** (Transformer with diagonal weights). A transformer with $L$ layers and $H$ attention heads on each layer is defined inductively by $\boldsymbol{Z}_0 = \boldsymbol{X} \in \mathbb{R}^{n \times d}$ and

- (Attention layer) $\tilde{\boldsymbol{Z}}_\ell = \boldsymbol{Z}_{\ell-1} + \sum_{i=1}^{H} \mathrm{attention}(\boldsymbol{Z}_{\ell-1}; \boldsymbol{W}_K^{\ell,i}, \boldsymbol{W}_Q^{\ell,i}, \boldsymbol{W}_V^{\ell,i}, \boldsymbol{W}_O^{\ell,i})$

- (Feedforward layer) $\boldsymbol{Z}_\ell = \tilde{\boldsymbol{Z}}_\ell + \sigma(\tilde{\boldsymbol{Z}}_\ell \boldsymbol{W}_A^\ell)(\boldsymbol{W}_B^\ell)^\top$ ,

where $\boldsymbol{W}_K^{\ell,i}, \boldsymbol{W}_Q^{\ell,i}, \boldsymbol{W}_V^{\ell,i}, \boldsymbol{W}_O^{\ell,i} \in \mathbb{R}^{d \times d'}$ are attention parameters, and $\boldsymbol{W}_A^\ell, \boldsymbol{W}_B^\ell \in \mathbb{R}^{d \times d'}$ are the feedforward parameters, and $\sigma$ is a continuously twice-differentiable activation. Suppose that the attention parameters are diagonal matrices: i.e., $\boldsymbol{W}_K^{\ell,i} = \mathrm{diag}(\boldsymbol{w}_K^{\ell,i}) \in \mathbb{R}^{d \times d}$, and similarly for the $\boldsymbol{W}_Q^{\ell,i}, \boldsymbol{W}_V^{\ell,i}, \boldsymbol{W}_O^{\ell,i}$ matrices. Then by the definition of the attention layer (1), the final output of the transformer $\boldsymbol{Z}_L$ only depends on the attention parameters through the elementwise products $\boldsymbol{w}_K^{\ell,i} \odot \boldsymbol{w}_Q^{\ell,i}$ and $\boldsymbol{w}_V^{\ell,i} \odot \boldsymbol{w}_O^{\ell,i}$. In other words, we can write

$$\boldsymbol{Z}_L = h(\boldsymbol{X}; \boldsymbol{u} \odot \boldsymbol{v}) \,,$$

for vectors $\boldsymbol{u} = [\boldsymbol{w}_K^{\ell,i}, \boldsymbol{w}_V^{\ell,i}]_{(\ell,i) \in [L] \times [H]} \in \mathbb{R}^{2dHL}$ and $\boldsymbol{v} = [\boldsymbol{w}_Q^{\ell,i}, \boldsymbol{w}_O^{\ell,i}]_{(\ell,i) \in [L] \times [H]} \in \mathbb{R}^{2dHL}$, and some smooth model $h$. Thus, if only the attention layers are trained, the diagonal transformer fits under Definition 3.1.

# 4  Incremental learning in networks with diagonal weights

We prove that if the initialization scale $\alpha$ is small, then learning proceeds in incremental stages, where in each stage the effective sparsity of the weights increases by at most one. These stages are implicitly defined by Algorithm 1 below, which constructs a sequence of times $0 = T_0 < T_1 < \cdots < T_k < \cdots$ and weight vectors $\boldsymbol{\theta}^0, \boldsymbol{\theta}^1, \ldots, \boldsymbol{\theta}^k, \ldots \in \mathbb{R}^{2p}$ that define the stages. We prove the following:

**Theorem 4.1** (Incremental dynamics at small initialization). *Let $f_{\mathsf{NN}}$ be a model with diagonal weights as in Definition 3.1. For any stage $k$ and time $t \in (T_k, T_{k+1})$ the following holds under Assumptions 2.1, 4.3, 4.4 and 4.5. There is $\alpha_0(t) > 0$ such that for all $\alpha < \alpha_0$, there exists a unique solution $\boldsymbol{\theta} : [0, t \log(1/\alpha)] \to \mathbb{R}^p$ to the gradient flow (3) and*

$$\lim_{\alpha \to 0} \boldsymbol{\theta}(t \cdot \log(1/\alpha)) \to \boldsymbol{\theta}^k \,,$$

*and at each stage the sparsity increases by at most one:* $\mathrm{supp}(\boldsymbol{\theta}^{k+1}) \setminus \mathrm{supp}(\boldsymbol{\theta}^k) \subseteq \{i_k\}$.[4]

**Application: transformer with diagonal weights**  Before giving the intuition for this theorem and stating the assumptions formally, let us discuss its application to the diagonal transformer model from Example 3.2. As a corollary of Theorem 4.1, the gradient flow on a diagonal transformer with small initialization will learn in stages, where in each stage there will be at most one head $i \in [H]$ in one layer $\ell \in [L]$ such that either the rank of $\boldsymbol{W}_K^{\ell,i}(\boldsymbol{W}_Q^{\ell,i})^\top = \mathrm{diag}(\boldsymbol{w}_K^{\ell,i})\mathrm{diag}(\boldsymbol{w}_Q^{\ell,i})$ or the rank of $\boldsymbol{W}_V^{\ell,i}(\boldsymbol{W}_O^{\ell,i})^\top = \mathrm{diag}(\boldsymbol{w}_V^{\ell,i})\mathrm{diag}(\boldsymbol{w}_O^{\ell,i})$ increases by at most one. In Figure 2, we illustrate these dynamics in the toy case of a single attention head trained in a student-teacher setup.

---

[4]Abusing notation, for $\boldsymbol{\theta} = (\boldsymbol{u}, \boldsymbol{v}) \in \mathbb{R}^p \times \mathbb{R}^p$, we write $\mathrm{supp}(\boldsymbol{\theta}) = \mathrm{supp}(\boldsymbol{u}) \cup \mathrm{supp}(\boldsymbol{v})$.

**Algorithm 1** Incremental learning in networks with diagonal weights

1: $\boldsymbol{b}^0, \boldsymbol{\theta}^0 \leftarrow \mathbf{0} \in \mathbb{R}^p, T_0 \leftarrow 0$
2: **for** stage number $k = 0, 1, 2, \ldots$ **do**
3:    # (A) Pick new coordinate $i_k \in [p]$ to activate.
4:    For each $i$, define time $\Delta_k(i)$ until active using (10).
5:    Pick winning coordinate $i_k$ using (11)
6:    Calculate time $T_{k+1}$ using (11) and **break** if $\infty$
7:    Update logarithmic weight approximation $\boldsymbol{b}^{k+1}$ using (12)
8:    # (B) Train activated coordinates to stationarity.
9:    $\boldsymbol{\theta}^{k+1} \leftarrow$ limiting dynamics point from (13)
10: **end for**

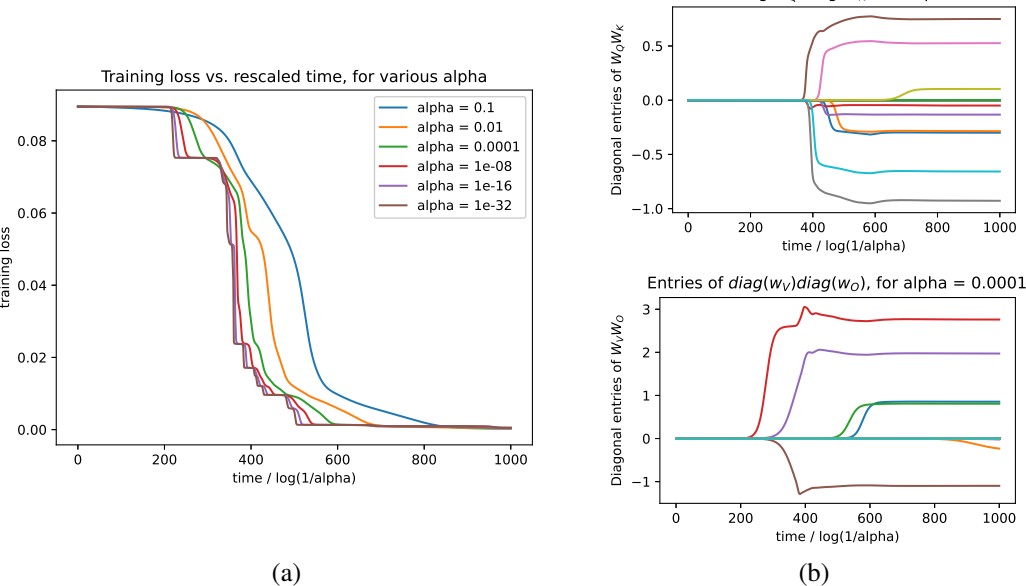

Figure 2: (a) Loss versus rescaled time in the toy task of learning an attention head with diagonal weights, for various initialization scales $\alpha$. The loss curves converge as $\alpha \to 0$ to a curve with stagewise loss plateaus and sharp decreases, as predicted by the theory; some stagewise learning behavior is already clear with $\alpha = 0.01$. (b) Each line shows the evolution of one of the entries of $\text{diag}(\boldsymbol{w}_Q)\text{diag}(\boldsymbol{w}_K)$ and $\text{diag}(\boldsymbol{w}_V)\text{diag}(\boldsymbol{w}_O)$ over rescaled time, demonstrating that the rank of these matrices increases incrementally; see Appendix A for experimental details and further results.

## 4.1 Intuition for incremental learning dynamics

We develop an informal intuition for Theorem 4.1 and fill out the definition of Algorithm 1. A model $f_{\text{NN}}$ with diagonal weights $\boldsymbol{\theta} = (\boldsymbol{u}, \boldsymbol{v})$ as in Definition 3.1 evolves under the gradient flow (3) as

$$\frac{d\boldsymbol{u}}{dt} = \boldsymbol{v} \odot \boldsymbol{g}(\boldsymbol{\theta}), \quad \frac{d\boldsymbol{v}}{dt} = \boldsymbol{u} \odot \boldsymbol{g}(\boldsymbol{\theta}) \quad \text{where} \tag{4}$$

$$\boldsymbol{g}(\boldsymbol{\theta}) = -\mathbb{E}_{\boldsymbol{x},\boldsymbol{y}}[D\ell(\boldsymbol{y}, h(\boldsymbol{x}; \boldsymbol{u} \odot \boldsymbol{v}))^\top Dh(\boldsymbol{x}; \boldsymbol{u} \odot \boldsymbol{v})^\top].$$

Here $D\ell(\boldsymbol{y}, \cdot) \in \mathbb{R}^{1 \times d_{out}}$ is the derivative of $\ell$ in the second argument and $Dh(\boldsymbol{x}, \cdot) \in \mathbb{R}^{d_{out} \times p}$ is the derivative of $h$ in the second argument. The first key observation is a conservation law that simplifies the dynamics. It can be viewed as the balancedness property for networks with linear activations [ACH18, DHL18], specialized to the case of diagonal layers.

**Lemma 4.2** (Conservation law). *For any $i \in [p]$ and any time $t$, we have*

$$u_i^2(t) - v_i^2(t) = u_i^2(0) - v_i^2(0). \tag{5}$$

*Proof.* This follows from $\frac{d}{dt}(u_i^2 - v_i^2) = u_i v_i g_i(\boldsymbol{\theta}) - u_i v_i g_i(\boldsymbol{\theta}) = 0.$ $\square$

The conservation law reduces the degrees of freedom and means that we need only keep track of $p$ parameters in total. Specifically, if we define $w_i(t) := u_i(t) + v_i(t)$, then the vector $\boldsymbol{w}(t) = \boldsymbol{u}(t) + \boldsymbol{v}(t)$ evolves by

$$\frac{d\boldsymbol{w}}{dt} = \boldsymbol{w} \odot \boldsymbol{g}(\boldsymbol{\theta})\,. \tag{6}$$

Using the conservation law (5), we can keep track of the weights in terms of the initialization and $\boldsymbol{w}(t)$:

$$\boldsymbol{\theta}(t) = \left( \frac{1}{2}(\boldsymbol{w}(t) + \frac{\boldsymbol{u}^{\odot 2}(0) - \boldsymbol{v}^{\odot 2}(0)}{\boldsymbol{w}(t)}), \frac{1}{2}(\boldsymbol{w}(t) - \frac{\boldsymbol{u}^{\odot 2}(0) - \boldsymbol{v}^{\odot 2}(0)}{\boldsymbol{w}(t)}) \right) \tag{7}$$

Therefore it suffices to analyze the dynamics of $\boldsymbol{w}(t)$.

### 4.1.1 Stage 1 of dynamics

**Stage 1A of dynamics: loss plateau for time** $\Theta(\log(1/\alpha))$    At initialization, $\boldsymbol{\theta}(0) \approx \boldsymbol{0}$ because the weights are initialized on the order of $\alpha$ which is small. This motivates the approximation $\boldsymbol{g}(\boldsymbol{\theta}(t)) \approx \boldsymbol{g}(\boldsymbol{0})$, under which the dynamics solve to:

$$\boldsymbol{w}(t) \approx \boldsymbol{w}(0) \odot e^{\boldsymbol{g}(\boldsymbol{0})t}. \tag{8}$$

Of course, this approximation is valid only while the weights are still close to the small initialization. The approximation breaks once one of the entries of $\boldsymbol{\theta}(t)$ reaches constant size. By combining (7) and (8), this happens at time $t \approx T_1 \cdot \log(1/\alpha)$ for

$$T_1 = \min_{i \in [p]} 1/|g_i(\boldsymbol{0})|\,.$$

Until this time, the network remains close to its initialization, and so we observe a loss plateau.

**Stage 1B of dynamics: nonlinear dynamics for time** $O(1)$    Subsequently, the loss decreases nonlinearly during a $O(1)$ time-scale, which is vanishingly short relative to the time-scale of the loss plateau. To prove this, we make the non-degeneracy assumption that there is a unique coordinate $i_0$ such that $1/|g_{i_0}(\boldsymbol{0})| = T_1$. Under this assumption, in stage 1A all weights except for those at coordinate $i_0$ remain vanishingly small, on the order of $o_\alpha(1)$. Concretely, for any small $\epsilon > 0$, there is a time $\underline{t}_1(\epsilon) \approx T_1 \cdot \log(1/\alpha)$ and sign $s \in \{+1, -1\}$ such that[5]

$$u_{i_0}(\underline{t}_1) \approx \epsilon, v_{i_0}(\underline{t}_1) \approx s\epsilon \quad \text{and} \quad |u_i(\underline{t}_1)|, |v_i(\underline{t}_1)| = o_\alpha(1) \text{ for all } i \neq i_0.$$

Because all coordinates except for $i_0$ have vanishingly small $o_\alpha(1)$ weights after stage 1A, we may perform the following approximation of the dynamics. Zero out the weights at coordinates except for $i_0$, and consider the training dynamics starting at $\tilde{\boldsymbol{\theta}} = (\epsilon \boldsymbol{e}_{i_0}, s\epsilon \boldsymbol{e}_{i_0})$. After $O(1)$ time, we should expect these dynamics to approach a stationary point. Although the evolution is nonlinear, all entries remain zero except for the $i_0$ entries, so the stationary point is also sparse. Mathematically, there is a time $\bar{t}_1 = \underline{t}_1 + O(1) \approx T_1 \cdot \log(1/\alpha)$ such that

$$\boldsymbol{\theta}(\bar{t}_1) \approx (a\boldsymbol{e}_{i_0}, sa\boldsymbol{e}_{i_0}) := \boldsymbol{\theta}^1\,,$$

for some $a \in \mathbb{R}_{>0}$, where $\boldsymbol{\theta}^1$ is a stationary point of the loss.[6] Despite the nonlinearity of the dynamics, the approximation can be proved using Grönwall's inequality since $\bar{t}_1 - \underline{t}_1 = O(1)$ is a constant time-scale.

To summarize, we have argued that the network approximately reaches stationary point that is 1-sparse, where only the weights at coordinate $i_0$ are nonzero.

---

[5]Without loss of generality, we can ensure that at initialization $\boldsymbol{u}(0)$ and $\boldsymbol{u}(0) + \boldsymbol{v}(0)$ are nonnegative. This implies $\boldsymbol{u}(t)$ is nonnegative. The fact that $u_{i_0}$ and $v_{i_0}$ are roughly equal in magnitude but might differ in sign is due to the conservation law (5). See Appendix C.3 for details.

[6]The entries of $\boldsymbol{u}$ and $\boldsymbol{v}$ are close in magnitude (but may differ in sign) because of the conservation law (5).

### 4.1.2 Later stages

We inductively extend the argument to any number of stages $k$, where each stage has a $\Theta(\log(1/\alpha))$-time plateau, and then a $O(1)$-time nonlinear evolution, with the sparsity of the weights increasing by at most one. The argument to analyze multiple stages is analogous, but we must also keep track of the magnitude of the weights on the logarithmic scale, since these determine how much longer . Inductively on $k$, suppose that there is some $T_k \in \mathbb{R}, \boldsymbol{b}^k \in \mathbb{R}^p$ and $\boldsymbol{\theta}^k \in \mathbb{R}^{2p}$ and a time $\bar{t}_k \approx T_k \cdot \log(1/\alpha)$ such that

$$\log_\alpha(\boldsymbol{w}(\bar{t}_k)) \approx \boldsymbol{b}^k \text{ and } \boldsymbol{\theta}(\bar{t}_k) \approx \boldsymbol{\theta}^k,$$

where $\boldsymbol{\theta}^k$ is a stationary point of the loss. Our inductive step shows that there is $T_{k+1} \in \mathbb{R}$ such that during times $t \in (\bar{t}_k, T_{k+1} \cdot \log(1/\alpha) - \Omega(1))$ the weights remain close to the stationary point from the previous stage, i.e., $\boldsymbol{\theta}(t) \approx \boldsymbol{\theta}^k$. And at a time $\bar{t}_{k+1} \approx T_{k+1} \cdot \log(1/\alpha)$ we have

$$\log_\alpha(\boldsymbol{w}(\bar{t}_{k+1})) \approx \boldsymbol{b}^{k+1} \text{ and } \boldsymbol{\theta}(\bar{t}_{k+1}) \approx \boldsymbol{\theta}^{k+1},$$

where $\boldsymbol{\theta}^{k+1}$ and $\boldsymbol{b}^{k+1}$ are defined below (summarized in Algorithm 1). Most notably, $\boldsymbol{\theta}^{k+1}$ is a stationary point of the loss whose support grows by at most one compared to $\boldsymbol{\theta}^k$.

**Stage $(k+1)$A, loss plateau for time $\Theta(\log(1/\alpha))$** At the beginning of stage $k+1$, the weights are close to the stationary point $\boldsymbol{\theta}^k$, and so, similarly to stage 1A, linear dynamics are valid.

$$\boldsymbol{w}(t) \approx \boldsymbol{w}(\bar{t}_k) \odot e^{\boldsymbol{g}(\boldsymbol{\theta}^k)(t-\bar{t}_k)} . \tag{9}$$

Using the conservation law (7), we derive a "time until active" for each coordinate $i \in [p]$, which corresponds to the time for the weight at that coordinate to grow from $o_\alpha(1)$ to $\Theta(1)$ magnitude:

$$\Delta_k(i) = \begin{cases} (b_i^k - 1 + \text{sgn}(g_i(\boldsymbol{\theta}^k)))/g_i(\boldsymbol{\theta}^k), & \text{if } g_i(\boldsymbol{\theta}^k) \neq 0 \\ \infty, & \text{if } g_i(\boldsymbol{\theta}^k) = 0 \end{cases} \tag{10}$$

The linear dynamics approximation (9) breaks down at a time $t \approx T_{k+1} \cdot \log(1/\alpha)$, where

$$T_{k+1} = T_k + \Delta_k(i_k), \quad i_k = \arg\min_{i \in [p]} \Delta_k(i) , \tag{11}$$

which corresponds to the first time at the weights at a coordinate grow from $o_\alpha(1)$ to $\Theta(1)$ magnitude. And at times $t \approx T_{k+1} \cdot \log(1/\alpha)$, on the logarithmic scale $\boldsymbol{w}$ is given by

$$\log_\alpha(\boldsymbol{w}(t)) \approx \boldsymbol{b}^{k+1} := \boldsymbol{b}^k - \boldsymbol{g}(\boldsymbol{\theta}^k)\Delta_k(i_k) , \tag{12}$$

**Stage $(k+1)$B of dynamics: nonlinear dynamics for time $O(1)$** Subsequently, the weights evolve nonlinearly during $O(1)$ time. In a similar way to the analysis of Stage 1B, we show that at a time $\bar{t}_{k+1} = \underline{t}_{k+1} + O(1) \approx T_{k+1} \cdot \log(1/\alpha)$, we have

$$\boldsymbol{\theta}(\bar{t}_{k+1}) \approx \boldsymbol{\theta}^{k+1} := \lim_{\epsilon \to 0} \lim_{t \to \infty} \boldsymbol{\psi}^k(t, \epsilon) , \tag{13}$$

where the dynamics $\boldsymbol{\psi}^k(t, \epsilon) \in \mathbb{R}^{2p}$ are initialized at $\boldsymbol{\psi}^k(0, \epsilon) = \boldsymbol{\theta}^k + (\epsilon \boldsymbol{e}_{i_k}, \text{sgn}(g_i(\boldsymbol{\theta}^k))\epsilon \boldsymbol{e}_{i_k})$ and evolve according to the gradient flow $\frac{d\boldsymbol{\psi}^k(t,\epsilon)}{dt} = -\nabla_{\boldsymbol{\theta}} \mathcal{L}(\boldsymbol{\psi}^k)$. This concludes the inductive step.

### 4.2 Assumptions for incremental dynamics

To make this intuition rigorous, we formalize below the assumptions required for Theorem 4.1. In Figure 3 and Appendix A, we provide experiments validating these assumptions on the toy model.

The first assumption is that the dynamics are non-degenerate, in the sense that two coordinates do not have weights that grow from $o_\alpha(1)$ to $\Theta(1)$ size at the same rescaled time. We also place a technical condition to handle the corner case when a coordinate leaves the support of the current stage's stationary point.

**Assumption 4.3** (Nondegeneracy of dynamics in part (A))**.** The initialization satisfies $|u_i(0)| \neq |v_i(0)|$ for all $i$. For stage $k$, either $T_{k+1} = \infty$ or there is a unique minimizer $i_k$ to $\min_i \Delta_k(i_k)$ in (11). Finally, for all $i \in \text{supp}(\boldsymbol{\theta}^{k-1}) \setminus \text{supp}(\boldsymbol{\theta}^k)$ we have $g_i(\boldsymbol{\theta}^k) \neq 0$.

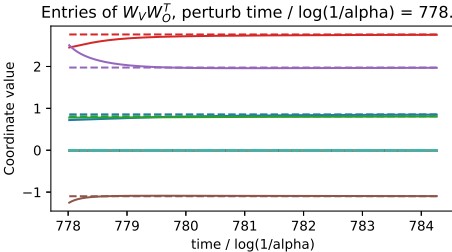 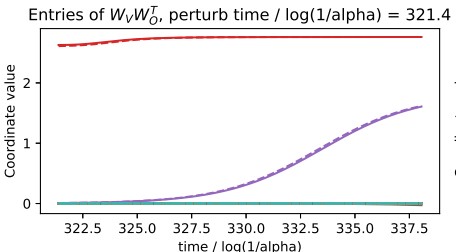

Figure 3: Validation of assumptions on the toy model of learning a single attention head. (a) Assumption 4.4: weights perturbed at a random time during training (solid lines) tend back to the near-stationary point (dashed lines). (b) Assumption 4.5: weights perturbed at the beginning of a stage (solid lines) have same nonlinear evolution as without perturbation (dashed lines). Details of these experiments and further validations are provided in Appendix A.

Next, we require that very small perturbations of the coordinates outside of $\mathrm{supp}(\boldsymbol{\theta}^k)$ do not change the dynamics. For this, it suffices that $\boldsymbol{\theta}^k$ be a strict local minimum.

**Assumption 4.4** (Stationary points are strict local minima)**.** For stage $k$, there exist $\delta_k > 0$ and $c_k > 0$ such that for $\tilde{\boldsymbol{u}} \in B(\boldsymbol{u}^k, \delta)$ supported on $\mathrm{supp}(\boldsymbol{u}^k)$, we have

$$\mathcal{L}(\tilde{\boldsymbol{u}}, \boldsymbol{s}^k \odot \tilde{\boldsymbol{u}}) \geq c_k \|\boldsymbol{u}^k - \tilde{\boldsymbol{u}}\|^2$$

Finally, we require a robust version of the assumption that the limit (13) exists, asking for convergence to a neighborhood of $\boldsymbol{\theta}^{k+1}$ even when the initialization is slightly noisy.

**Assumption 4.5** (Noise-robustness of dynamics in part (B))**.** For any stage $k$ with $T_{k+1} < \infty$ and any $\epsilon > 0$, there are $\delta > 0$ and $\tau : \mathbb{R}_{>0} \to \mathbb{R}$ such that the following holds. For any $\tilde{\boldsymbol{u}} \in B(\boldsymbol{u}^k, \delta) \cap \mathbb{R}_{\geq 0}^p$ supported on $\mathrm{supp}(\tilde{\boldsymbol{u}}) \subseteq \mathrm{supp}(\boldsymbol{u}^k) \cup \{i_k\}$, there exists a unique solution $\boldsymbol{\psi} : [0, \infty) \to \mathbb{R}^p$ of the gradient flow $\frac{d\boldsymbol{\psi}}{dt} = -\nabla_{\boldsymbol{\theta}} \mathcal{L}(\boldsymbol{\psi})$ initialized at $\boldsymbol{\psi}(0) = (\tilde{\boldsymbol{u}}, \boldsymbol{s}^{k+1} \odot \tilde{\boldsymbol{u}})$, and at times $t \geq \tau(\tilde{\psi}_{i_k})$,

$$\|\boldsymbol{\psi}(t) - \boldsymbol{\theta}^{k+1}\| < \epsilon \,.$$

## 5 Experimental results

We run experiments that go beyond the toy diagonal attention head model (see Figures 2 and 3) to test the extent to which low-rank incremental learning occurs in popular models used in practice. We conduct experiments with vision transformers (ViT) [DBK+20] trained on the CIFAR-10/100 and ImageNet datasets, and with the GPT-2 language transformer [BMR+20] trained on the Wikitext-103 dataset. Full experiments are deferred to Appendix B.

**Gradual rank increase in vision and language models** We train practical transformer architectures on vision and language tasks using Adam and the cross-entropy loss. We train all layers (including the feedforward layers). To capture the low-rank bias with a non-vanishing initialization scale, we study the spectrum of the difference $\Delta \boldsymbol{W}_K \boldsymbol{W}_Q^\top$ and $\Delta \boldsymbol{W}_V \boldsymbol{W}_O^\top$ between the weights post-training and their initial values. Specifically, in Figure 4, we plot the stable rank of the differences $\Delta \boldsymbol{W}_K \boldsymbol{W}_Q^\top$ and $\Delta \boldsymbol{W}_V \boldsymbol{W}_O^\top$. The weight perturbation learned during the training process gradually increases in stable rank during training, and is ultimately low-rank when compared to the initial spectrum. Finally, for CIFAR-10, we plot the spectrum of $\Delta \boldsymbol{W}_K \boldsymbol{W}_Q^\top$ against that of its initialized state in Figure 5 for different self-attention heads, illustrating that the weight perturbation learned during the training process is extremely low-rank when compared to the initial spectrum. In Appendix B, we also study optimization with SGD, which shows similar gradual rank increase behavior.

**Effect of initialization scale** We probe the effect of initialization scale on gradual rank increase dynamics for a ViT trained on CIFAR-10. We use a ViT of depth 6, with 8 self-attention heads per layer (with layer normalization). We use an embedding and MLP dimension of $d_{\mathrm{emb}} = 512$, and a head dimension of $d_h = 128$ (i.e $\boldsymbol{W}_K, \boldsymbol{W}_Q, \boldsymbol{W}_V, \boldsymbol{W}_O \in \mathbb{R}^{d_{\mathrm{emb}} \times d_h}$). We train the transformer

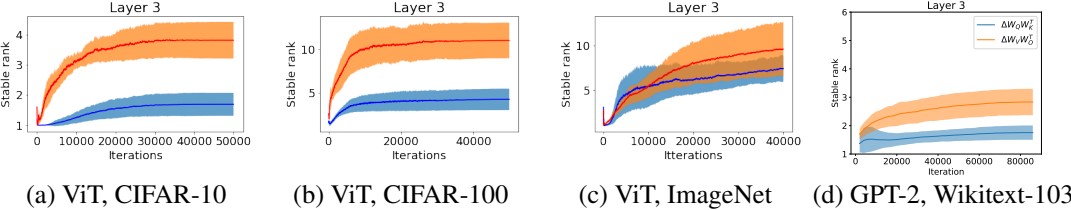

(a) ViT, CIFAR-10     (b) ViT, CIFAR-100     (c) ViT, ImageNet     (d) GPT-2, Wikitext-103

Figure 4: Stable rank of $\Delta \boldsymbol{W}_K \boldsymbol{W}_Q^\top$ (blue) and $\Delta \boldsymbol{W}_V \boldsymbol{W}_O^\top$ (orange) on an arbitrary chosen layer throughout training for four different pairs of networks and tasks. The stable rank of a matrix $\boldsymbol{W}$ is defined as $\|\boldsymbol{W}\|_F^2 / \|\boldsymbol{W}\|_2^2$, and gives a smooth approximation of the rank. Mean and standard deviation (shaded area) are computed across all heads in each attention layer. Full details and results are in Appendix B.

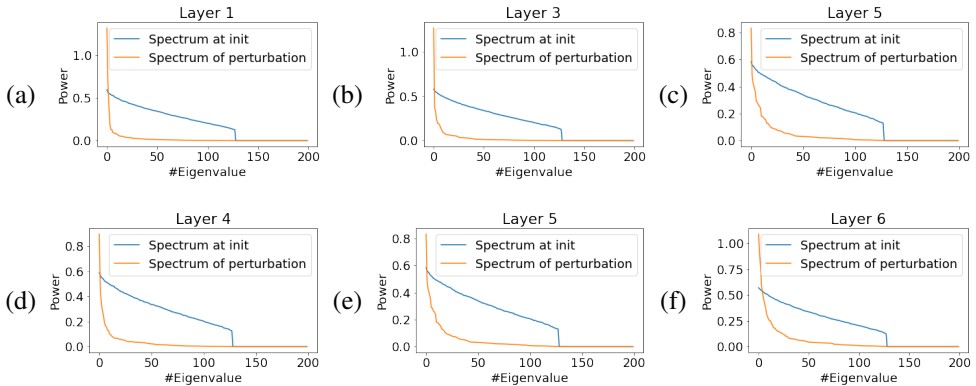

Figure 5: Spectrum of the weight perturbation $\Delta \boldsymbol{W}_K \boldsymbol{W}_Q^\top$ vs. initialization in a vision transformer trained on CIFAR-10, using Adam and default initialization scale, in random self-attention heads in different layers. The learned perturbation exhibits extreme low-rank bias post-training even in default initialization scales. Analogous plots for CIFAR-100 and ImageNet are in Appendix B.

using Adam with the cross-entropy loss. We train all layers (including the feedforward layers) while varying the initialization scale of all layers by multiplying their initial values by a scale factor (we fix the scale of the initial token mapper). Figure 6 shows the evolution of the principal components of $\Delta \boldsymbol{W}_K \boldsymbol{W}_Q^\top$ and $\Delta \boldsymbol{W}_V \boldsymbol{W}_O^\top$ for a randomly-chosen self-attention head and layer throughout training, exhibiting incremental learning dynamics and a low-rank bias. Note that incremental learning and low-rank bias are increasingly evident with smaller initialization scales, as further demonstrated in Figure 7.

## 6 Discussion

We have identified incremental learning dynamics in transformers, proved them rigorously in a simplified setting, and shown them experimentally in networks trained with practical hyperparameters.

**Limitations** There are clear limitations to our theory: the diagonal weights and small initialization assumptions. More subtly, the theory does not apply to losses with exponential-like tails because the weights may not converge to a finite value and so Assumption 4.4 is not met (this could possibly be addressed by adding regularization). Also, the architecture must be smooth, which precludes ReLUs – but allows for smoothed ReLUs such as the GeLUs used in ViT [DBK+20]. Finally, the theory is for training with gradient flow, while other optimizers such as Adam are used in practice instead [KB14]. Nevertheless, our experiments on ViTs indicate that the incremental learning dynamics occur even when training with Adam.

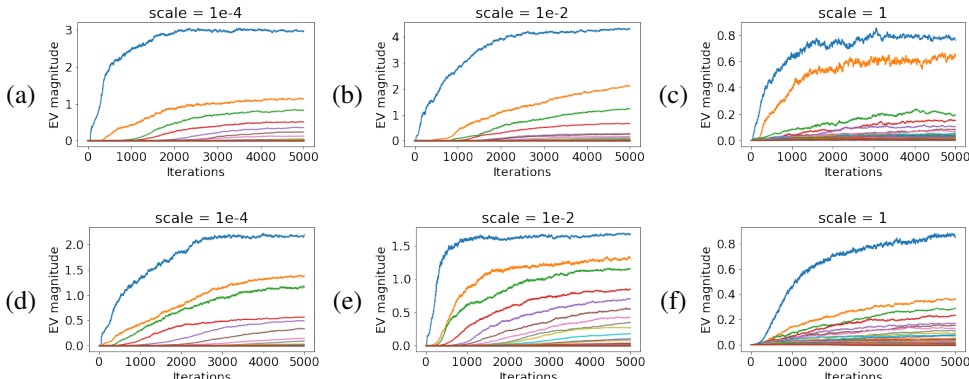

Figure 6: Training a vision transformer on CIFAR-10 using Adam, while varying the initialization scale (unit scale indicates default initialization). Plotted are the evolution of the eigenvalues of $\Delta \boldsymbol{W}_K \boldsymbol{W}_Q^\top$ (a) - (c) and $\Delta \boldsymbol{W}_V \boldsymbol{W}_O^\top$ (d) - (f) in a random self-attention head in the second layer throughout training. Incremental learning dynamics and a low-rank bias are evident for all scales, albeit more pronounced at smaller initialization scales.

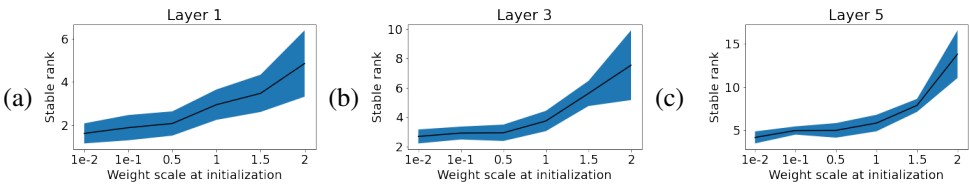

Figure 7: Stable rank of $\Delta \boldsymbol{W}_K \boldsymbol{W}_Q^\top$ per initialization scale (Unit scale refers to the default initialization) in different self-attention heads post-training, at layers 1, 3, 5. At each layer, the stable rank mean and standard deviation are computed across 8 heads per layer, for each initialization scale. All models were trained on CIFAR-10 using the Adam optimizer. Smaller initialization scales lead to lower-rank attention heads.

**Future directions**  An interesting avenue of future research is to develop a theoretical understanding of the implicit bias in function space of transformers whose weights are a low-rank perturbation of randomly initialized weights. Another promising direction is to examine the connection between our results on incremental dynamics and the LoRA method [HSW+21], with the goal of explaining and improving on this algorithm; see also the discussion in Section 1.1. Along this vein, a concurrent work [ZZC+23] independently observes gradual rank increase dynamics during transformer training and this inspires a low-rank training algorithm that obtains runtime and memory improvements over regular training. The results of [ZZC+23] are complementary to ours, since they study the feedforward layers of the transformer, and their theory applies to *linear* networks in the standard initialization scale; in contrast, we study the attention layers, and our theory applies to *nonlinear* networks with small initialization scale.

## Acknowledgments

We would like to thank Vimal Thilak for his help in setting up the infrastructure for conducting experiments, and the anonymous reviewers for their helpful feedback.

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

# Contents

# A  Experimental validation of the assumptions in Theorem 4.1

In Figures 2, 8, and 9, we plot the evolution of the losses, of the entries of $\boldsymbol{W}_K \boldsymbol{W}_Q^\top = \mathrm{diag}(\boldsymbol{w}_K)\mathrm{diag}(\boldsymbol{w}_Q)$, and of the entries of $\boldsymbol{W}_V \boldsymbol{W}_O^\top = \mathrm{diag}(\boldsymbol{w}_V)\mathrm{diag}(\boldsymbol{w}_O)$ in the toy task of training an attention head (1) with diagonal weights. The model is trained with SGD on the mean-squared error loss on 1000 random samples $(\boldsymbol{X}, \boldsymbol{y})$. Each random sample has $\boldsymbol{X} \in \mathbb{R}^{10 \times 50}$, which a sequence of 10 tokens, each of dimension 50, which is distributed as isotropic Gaussian. The label $\boldsymbol{y}$ is given by a randomly-generated teacher model that is also an attention head (1) with diagonal weights. In Figures 2, 8, and 9, for $\alpha \in \{0.1, 0.01, 0.0001, 10^{-8}, 10^{-16}, 10^{-32}\}$ we plot the evolution of the loss and of the weights when initialized at $\boldsymbol{\theta}(0) = \alpha \boldsymbol{\theta}_0$, for some random Gaussian $\boldsymbol{\theta}_0$. Qualitatively, as $\alpha \to 0$ we observe that the loss curve and the trajectories of the weights appear to converge to a limiting stagewise dynamics, where there are plateaus followed by movement on short time-scales, as predicted by the theory.

**Validation of Assumption 4.3 (non-degeneracy of dynamics)**  As $\alpha \to 0$, notice that the stages appear to separate and happen at distinct times. Furthermore, the extra technical condition on coordinates $i \in \mathrm{supp}(\boldsymbol{\theta}^k) \setminus \mathrm{supp}(\boldsymbol{\theta}^{k-1})$ in Assumption 4.3 is satisfied since no coordinates ever leave the support of $\boldsymbol{\theta}^k$.

**Validation of Assumption 4.4 (stationary points are strict local minima)**  In Figure 10 we consider the $\alpha = 10^{-32}$ trajectory, since this is closest to the dynamics in the $\alpha \to 0$ limit. We randomly select several epochs. Since the transitions between stages are a vanishing fraction of the total training time, each of these randomly-selected epochs is likely during a plateau, as we see in the figure. For each epoch perform the following experiment. For each nonnegligible coordinate of the weights (those where the weight is of magnitude greater than the threshold $\tau = 10^{-5}$), we perturb the weights by adding noise of standard deviation $0.05$. We then run the training dynamics starting at this perturbed initialization for 1000 epochs. We observe that the training dynamics quickly converge to the original unperturbed initialization, indicating that the weights were close to a strict local minimum of the loss.

**Validation of Assumption 4.5 (noise-robustness of dynamics)**  In Figure 11 we perform the same experiment as in Figure 10, except that the epochs we select to perturb the weights are those where there is a newly-nonnegligible coordinate (less than $10^{-5}$ in magnitude in the previous epoch, and more than $10^{-5}$ in magnitude in this epoch). We find that the nonlinear dynamics are robust and tend to the limiting endpoint even under a random Gaussian perturbation of standard deviation $10^{-2}$ on each of the nonnegligible coordinates, supporting Assumption 4.5.

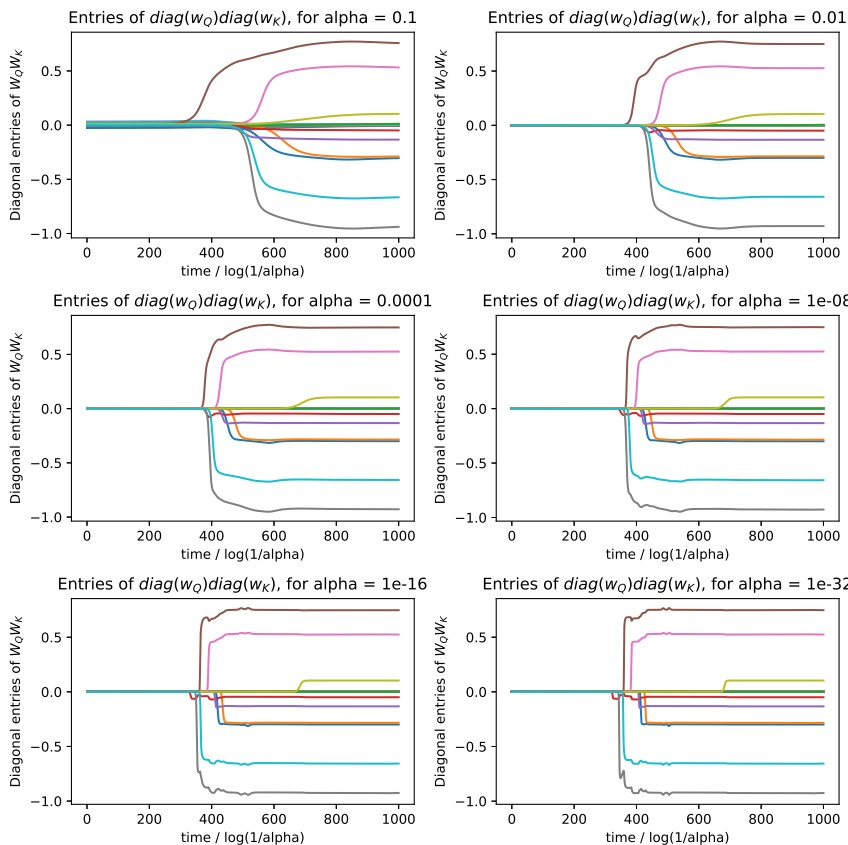

Figure 8: Evolution of $\mathrm{diag}(\boldsymbol{w}_Q)\mathrm{diag}(\boldsymbol{w}_K)$ entries over rescaled time initializing at various scalings $\alpha$. Notice that as $\alpha \to 0$, the training trajectories tend to a limiting trajectory. Each line corresponds to a diagonal entry of $\mathrm{diag}(\boldsymbol{w}_Q)\mathrm{diag}(\boldsymbol{w}_K)$.

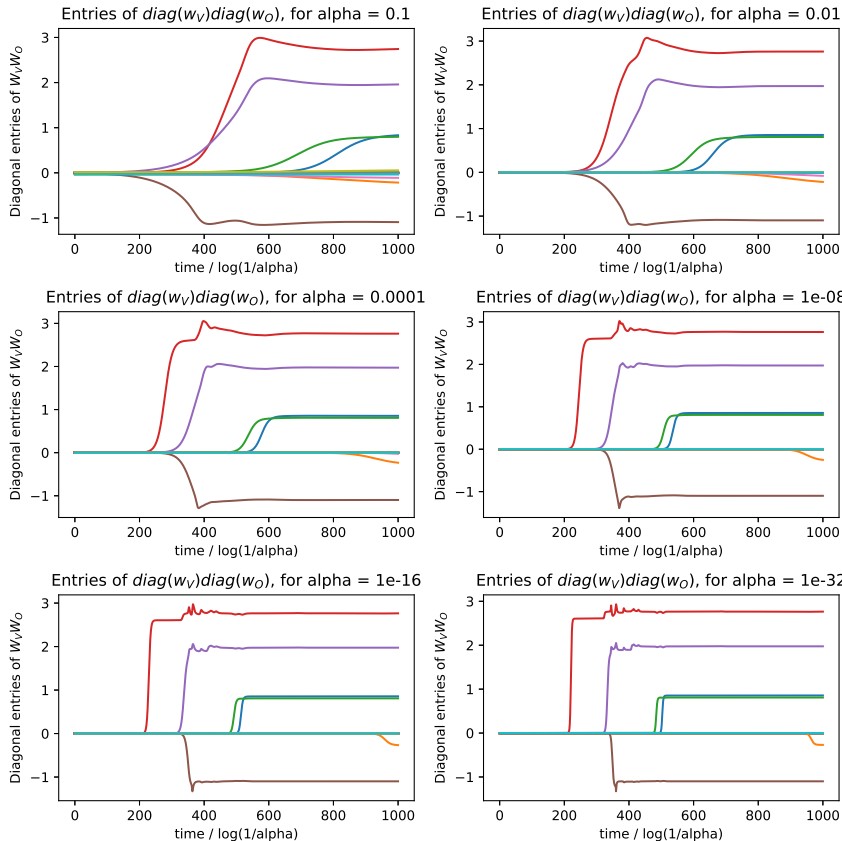

Figure 9: Evolution of $\mathrm{diag}(\boldsymbol{w}_V)\mathrm{diag}(\boldsymbol{w}_O)$ entries in the toy task of learning an attention head with diagonal weights. Each line corresponds to the evolution of an entry of $\mathrm{diag}(\boldsymbol{w}_V)\mathrm{diag}(\boldsymbol{w}_O)$ over rescaled time. Each plot corresponds to a different initialization magnitude $\alpha$. Notice that as $\alpha \to 0$, the training trajectories tend to a limiting trajectory.

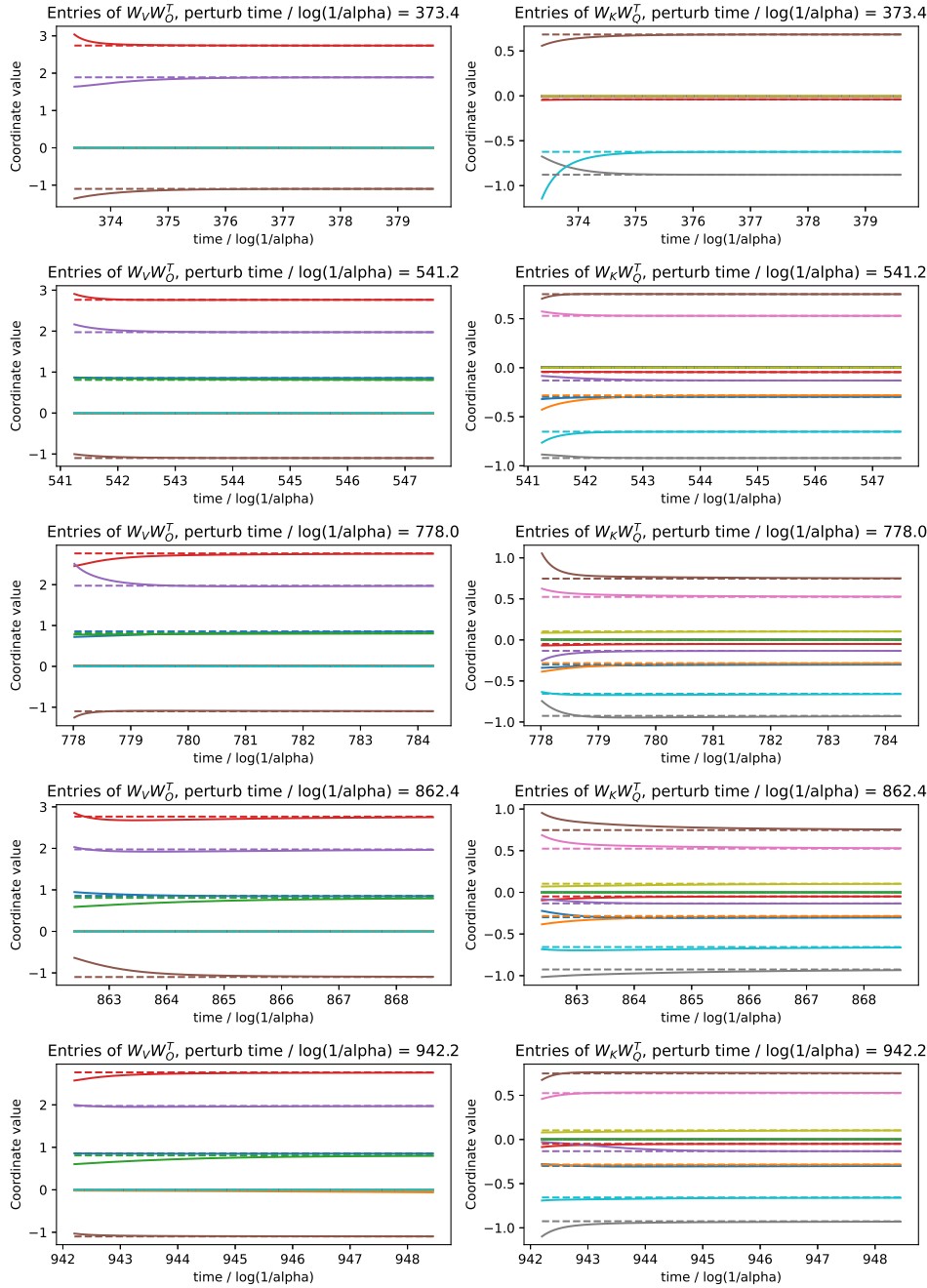

Figure 10: Evolution of weights of toy attention model under perturbation, validating Assumption 4.4. At 5 different random times during training, we perturb the nonnegligible weight coordinates and continue to train with SGD. The evolution of each of the weights under the initial perturbation (solid line) is compared to the original evolution without perturbation (dashed line). Observe that the training dynamics quickly brings each weight back to the unperturbed weight trajectory, indicating that the weights are originally close to a strict local minimum.

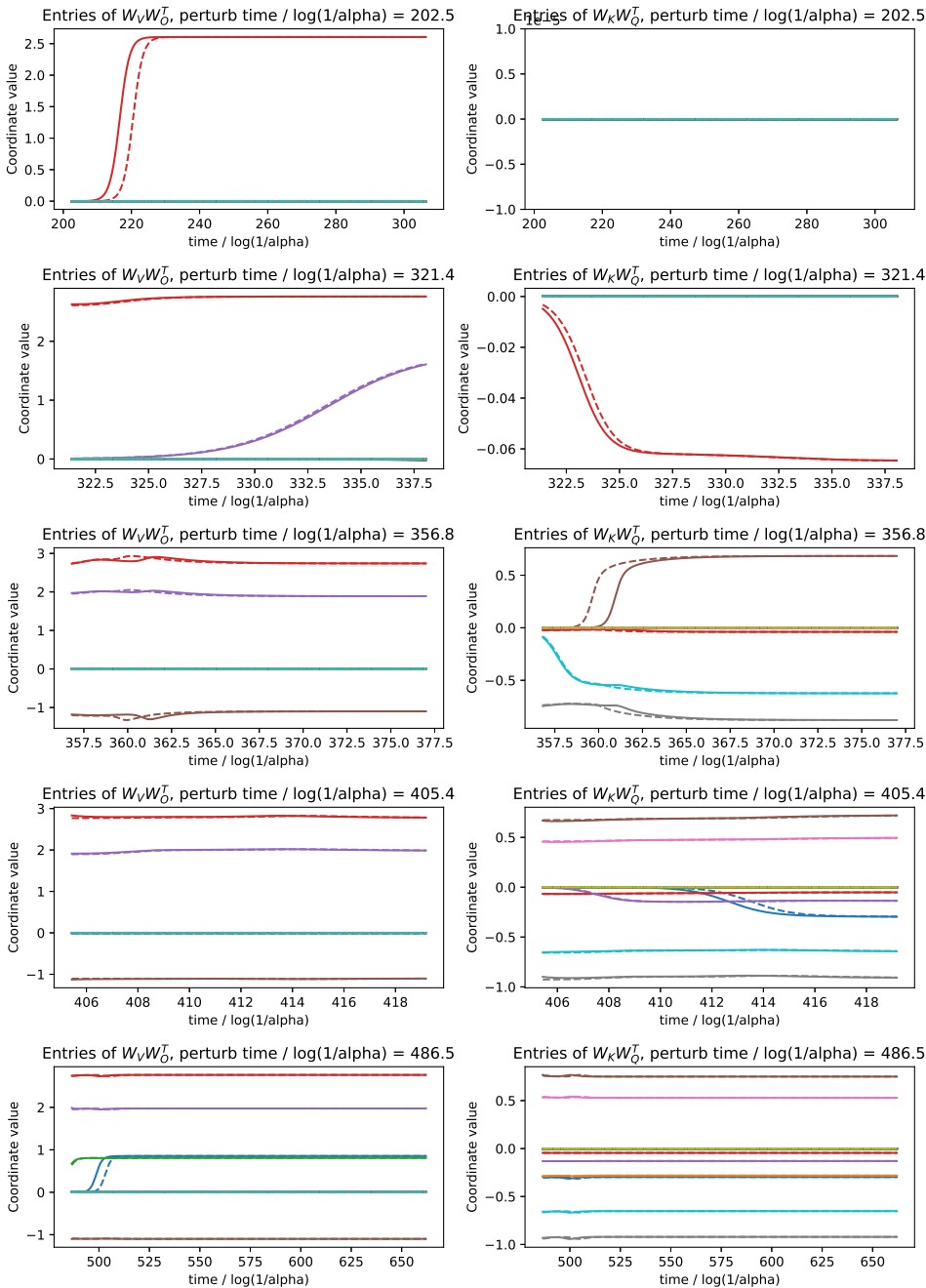

Figure 11: Validating Assumption 4.5 with the same experiment as in Figure 10, except that the epochs for the perturbation chosen are those where there is a newly nonnegligible coordinate. Perturbed dynamics (solid lines) are again robust to perturbation and track the original dynamics (dashed lines).

# B Further experiments on vision and language transformers

The practice of training transformer models often deviates substantially from the assumptions made in our theoretical analysis, and it is a priori unclear to what extent gradual rank increase behaviour and a low rank bias are manifested in setups more common in practical applications. To gauge the relevancy of our findings we conduct experiments on popular vision and language benchmarks, using algorithms and hyperparameters common in the literature. We use the stable rank of a matrix $\boldsymbol{W}$ given by $\frac{\|\boldsymbol{W}\|_F^2}{\|\boldsymbol{W}\|_2^2}$ as a smooth approximation of rank. We track the value of the stable rank for the different attention matrices throughout training. Although we do not expect our theoretical results to to hold precisely in practice, we find evidence of gradual increase in stable rank, leading to a low rank bias in Figures 12, 13, 15, 17 and 19. In these experiments we use off-the-shelf vision transformers (ViT) [DBK+20] trained on popular vision benchmarks, as well as off-the-shelf GPT-2 trained on a popular language benchmark. We use no weight decay or dropout in our experiments. All models were initialized using the default initialization scale.

## B.1 SGD-trained transformers

**CIFAR-10/100** We trained a 6-layer ViT with 8 heads per layer, embedding dimension 512, head dimension 128, and MLP dimension 512 and patch-size 4 for 500 epochs on CIFAR10/CIFAR100 with SGD and learning rate 3e-1 and warmup. See Figures 12 and 13. Each run took 2 hours on one A100 GPU.

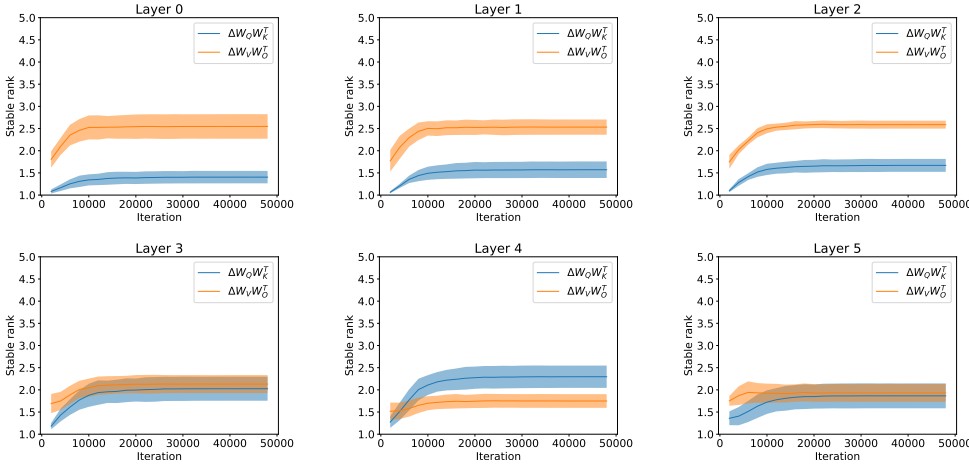

Figure 12: CIFAR-10, ViT trained with SGD: Stable rank of $\Delta\boldsymbol{W}_K\boldsymbol{W}_Q^\top$ (blue) and $\Delta\boldsymbol{W}_V\boldsymbol{W}_O^\top$ (orange) throughout training. Mean and standard deviation (shaded area) are computed across 8 heads per attention layer.

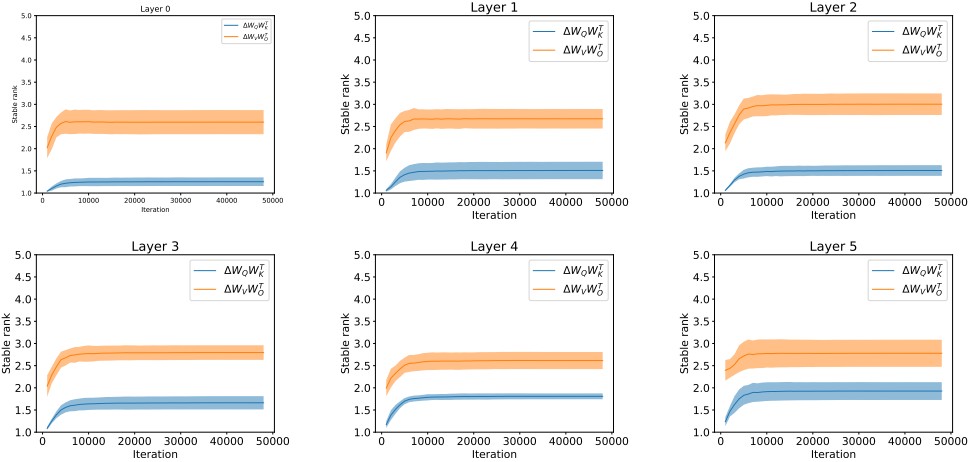

Figure 13: CIFAR-100, ViT trained with SGD: Stable rank of $\Delta \boldsymbol{W}_K \boldsymbol{W}_Q^\top$ (blue) and $\Delta \boldsymbol{W}_V \boldsymbol{W}_O^\top$ (orange) throughout training. Mean and standard deviation (shaded area) are computed across 8 heads per attention layer.

## B.2 Adam-trained transformers

**CIFAR-10/100** For the CIFAR-10/100 datasets we use a ViT with 6 layers, patchsize of 4, 8 heads per self attention layer, an embedding and MLP dimension of 512, and a head dimension of 128. We train the model using the Adam optimizer for 500 epochs with a base learning rate of 1e-4, a cyclic learning rate decay with a linear warmup schedule for 15 epochs and a batchsize of 512. Our results are summarized in Figures 14 and 15 for CIFAR-10, and Figures 16 and 17 for CIFAR-100.

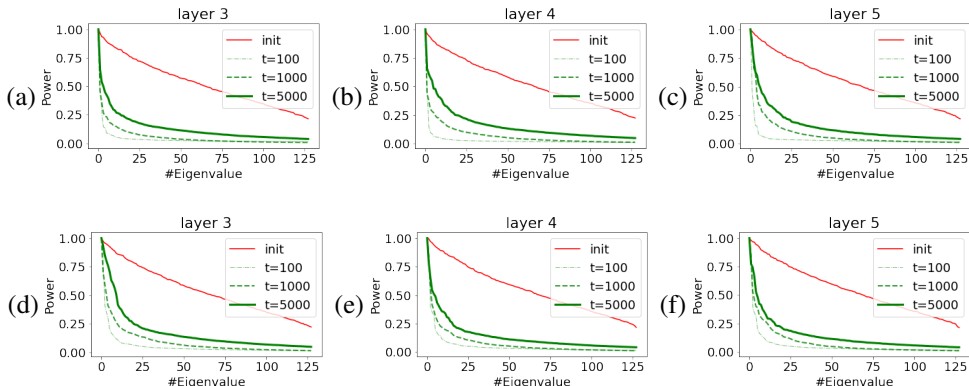

Figure 14: CIFAR-10, ViT trained with Adam: normalized spectrum at different stages of training. (a) - (c) Normalized spectrum of $\boldsymbol{W}_K \boldsymbol{W}_Q^\top$ at initialization and $\Delta \boldsymbol{W}_K \boldsymbol{W}_Q^\top$ during training for different attention heads at different layers. (d) - (e) equivalent figures for $\boldsymbol{W}_V \boldsymbol{W}_O^\top$.

**ImageNet** For ImageNet, we use the ViT-Base/16 from [DBK+20] trained with Adam for 360 epochs with a base learning rate of 3e-3, a cyclic learning rate decay with a linear warmup schedule for 15 epochs and a batchsize of 4096. Our results are summarized in Figures 18 and 19 for ImageNet.

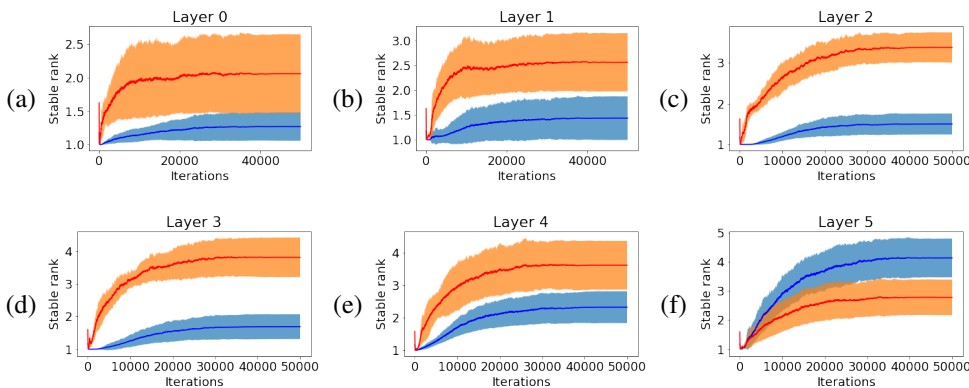

Figure 15: CIFAR-10, ViT trained with Adam: Stable rank of $\Delta \boldsymbol{W}_K \boldsymbol{W}_Q^\top$ (blue) and $\Delta \boldsymbol{W}_V \boldsymbol{W}_O^\top$ (red) throughout training. Mean and standard deviation (shaded area) are computed across 8 heads per attention layer.

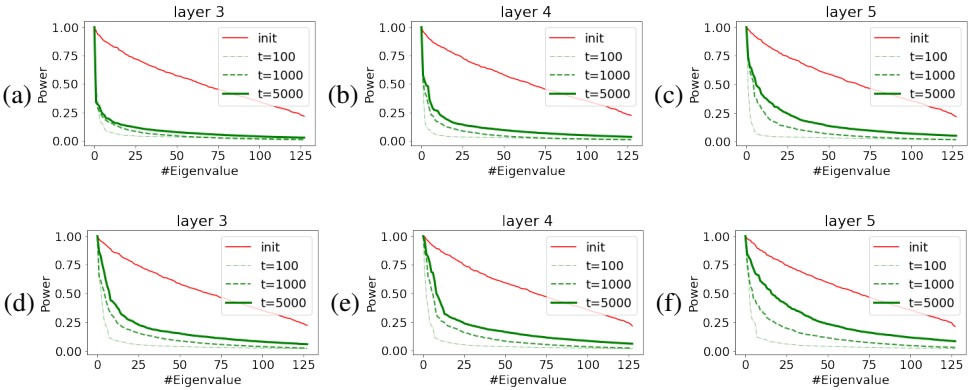

Figure 16: CIFAR-100, ViT trained with Adam: normalized spectrum at different stages of training. (a) - (c) Normalized spectrum of $\boldsymbol{W}_K \boldsymbol{W}_Q^\top$ at initialization and $\Delta \boldsymbol{W}_K \boldsymbol{W}_Q^\top$ during training for different attention heads at different layers. (d) - (e) equivalent figures for $\boldsymbol{W}_V \boldsymbol{W}_O^\top$.

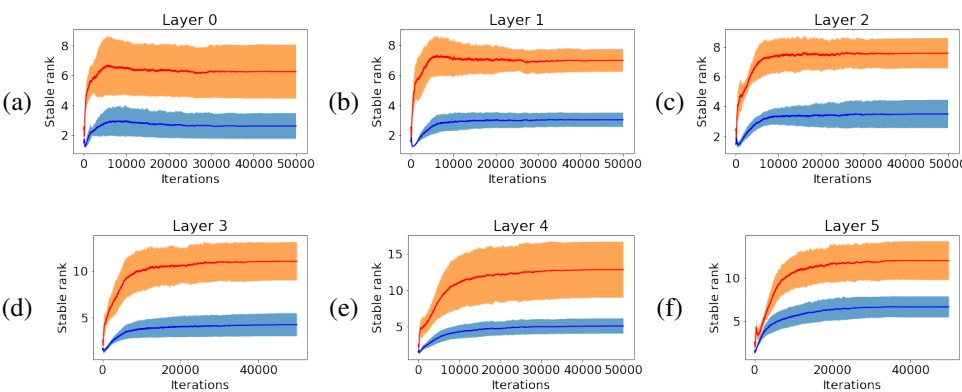

Figure 17: CIFAR-100, ViT trained with Adam: Stable rank of $\Delta \boldsymbol{W}_K \boldsymbol{W}_Q^\top$ (blue) and $\Delta \boldsymbol{W}_V \boldsymbol{W}_O^\top$ (red) throughout training. Mean and standard deviation (shaded area) are computed across 8 heads per attention layer.

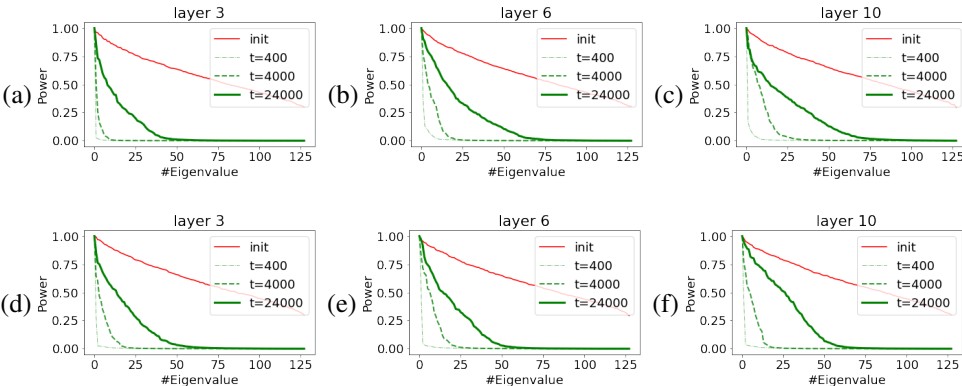

Figure 18: ImageNet, ViT trained with Adam: normalized spectrum at different stages of training. (a) - (c) Normalized spectrum of $\boldsymbol{W}_K \boldsymbol{W}_Q^\top$ at initialization and $\Delta \boldsymbol{W}_K \boldsymbol{W}_Q^\top$ during training for different attention heads at different layers. (d) - (e) equivalent figures for $\boldsymbol{W}_V \boldsymbol{W}_O^\top$.

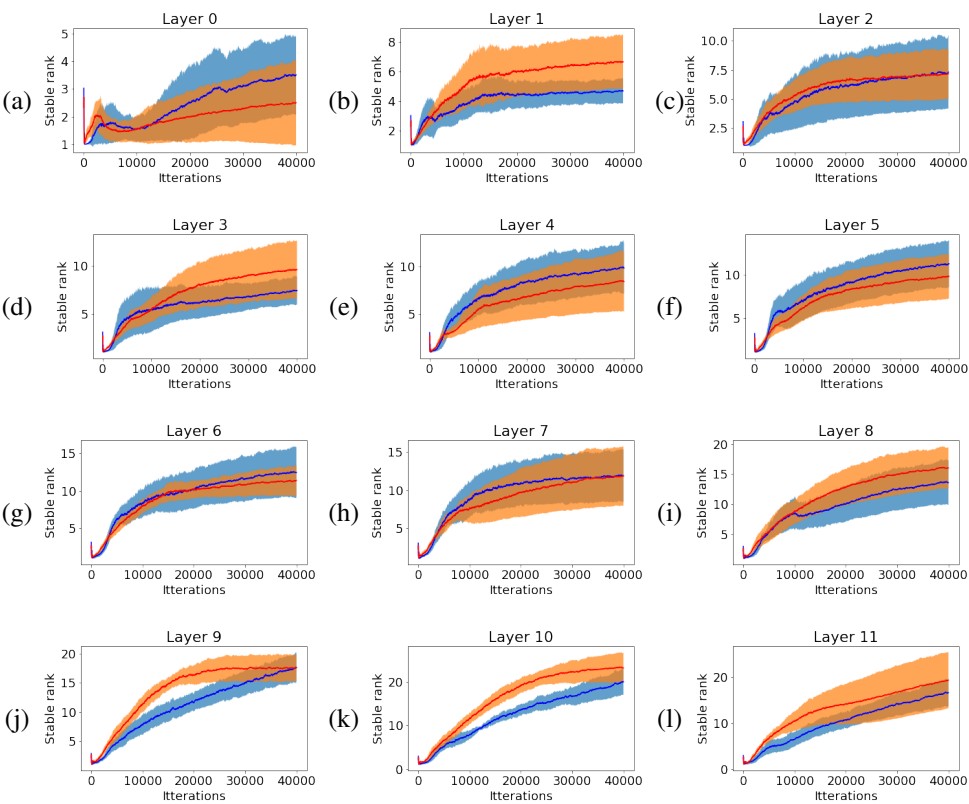

Figure 19: ImageNet, ViT trained with Adam: Stable rank of $\Delta \boldsymbol{W}_K \boldsymbol{W}_Q^\top$ (blue) and $\Delta \boldsymbol{W}_V \boldsymbol{W}_O^\top$ (red) throughout training. Mean and standard deviation (shaded area) are computed across 12 heads per attention layer.

**Wikitext-103**  The gradual rank increase phenomenon also occurs in the NLP setting with language transformers. We trained GPT-2 on Wikitext-103 using the HuggingFace training script with Adam learning rate 3e-4, per-GPU batch-size 8, and block-length 256. We trained for 3 epochs on 2 A100 GPUs, which took 12 hours. See Figure 20.

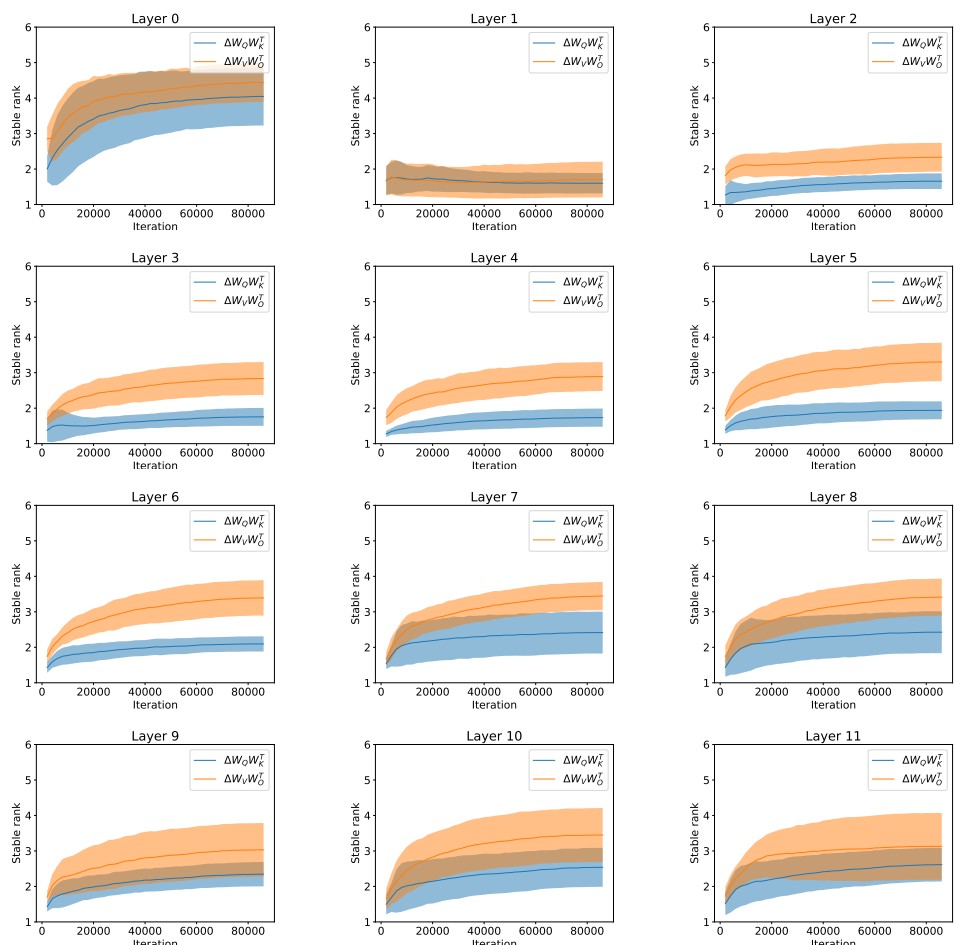

Figure 20: Wikitext-103, GPT-2 trained with Adam: Stable rank of $\Delta \boldsymbol{W}_V \boldsymbol{W}_O^\top$ and $\Delta \boldsymbol{W}_Q \boldsymbol{W}_K^\top$, versus training iteration. Stable rank of the perturbation increases gradually, but remains small throughout training.

# C Proof for dynamics of networks with diagonal parametrization (Theorem 4.1)

## C.1 Assumptions

Recall we have defined $\boldsymbol{\theta}^0, \ldots, \boldsymbol{\theta}^k, \ldots \in \mathbb{R}^{2p}$ as the sequence of weights such that $\boldsymbol{\theta}^0 = \mathbf{0}$ and $\boldsymbol{\theta}^{k+1}$ is defined inductively as follows. Consider the dynamics of $\boldsymbol{\psi}^k(t, \epsilon) \in \mathbb{R}^{2p}$ initialized at $\boldsymbol{\psi}^k(0, \epsilon) = \boldsymbol{\theta}^k + (\epsilon \boldsymbol{e}_{i_k}, \mathrm{sgn}(g_i(\boldsymbol{\theta}^k))\epsilon \boldsymbol{e}_{i_k})$ and evolving according to the gradient flow $\frac{d\boldsymbol{\psi}^k(t,\epsilon)}{dt} = -\nabla_{\boldsymbol{\theta}} \mathcal{L}(\boldsymbol{\psi}^k)$. We assume that there is a limiting point $\boldsymbol{\theta}^{k+1}$ of these dynamics as $\epsilon$ is taken small and the time is taken large:

$$\lim_{\epsilon \to 0} \lim_{t \to \infty} \boldsymbol{\psi}^k(t, \epsilon) = \boldsymbol{\theta}^{k+1} \,.$$

Under the above assumption that this sequence $\boldsymbol{\theta}^0, \ldots, \boldsymbol{\theta}^k, \ldots$ is well-defined, we can derive a useful property of it for free. Namely, the conservation law (5) implies that $\boldsymbol{u} \odot \boldsymbol{u} - \boldsymbol{v} \odot \boldsymbol{v}$ is preserved. It follows that for each $k$ we have that $\boldsymbol{\theta}^k = (\boldsymbol{u}^k, \boldsymbol{v}^k)$ satisfies $|\boldsymbol{u}^k| = |\boldsymbol{v}^k|$ entrywise. In other words, there is $\boldsymbol{s}^k \in \{+1, -1\}^p$ satisfying

$$\boldsymbol{\theta}^k = (\boldsymbol{u}^k, \boldsymbol{s}^k \odot \boldsymbol{u}^k) \in \mathbb{R}^{2p} \,.$$

We also abuse notation and write $\mathrm{supp}(\boldsymbol{\theta}^k) := \mathrm{supp}(\boldsymbol{u}^k) \subseteq [p]$, since the support of $\boldsymbol{\theta}^k$ on the first $p$ coordinates matches its support on the last $p$ coordinates.

Having fixed this notation, we now recall the main assumptions of the theorem.

**Assumption C.1** (Nondegeneracy of dynamics in part (A); Assumption 4.3). The initialization satisfies $|u_i(0)| \neq |v_i(0)|$ for all $i$. For stage $k$, either $T_{k+1} = \infty$ or there is a unique minimizer $i_k$ to $\min_i \Delta_k(i_k)$ in (11). Finally, for all $i \in \mathrm{supp}(\boldsymbol{\theta}^{k-1}) \setminus \mathrm{supp}(\boldsymbol{\theta}^k)$ we have $g_i(\boldsymbol{\theta}^k) \neq 0$.

**Assumption C.2** (Stationary points are strict local minima; Assumption 4.4). For stage $k$, there exist $\delta_k > 0$ and $c_k > 0$ such that for $\tilde{\boldsymbol{u}} \in B(\boldsymbol{u}^k, \delta)$ supported on $\mathrm{supp}(\boldsymbol{u}^k)$, we have

$$\mathcal{L}(\tilde{\boldsymbol{u}}, \boldsymbol{s}^k \odot \tilde{\boldsymbol{u}}) \geq c_k \|\boldsymbol{u}^k - \tilde{\boldsymbol{u}}\|^2 \,.$$

**Assumption C.3** (Noise-robustness of dynamics in part (B); Assumption 4.5). For stage $k$, either $T_{k+1} = \infty$ or the following holds. For any $\epsilon > 0$, there are $\delta > 0$ and $\tau : \mathbb{R}_{>0} \to \mathbb{R}$ such that the following holds. For any $\tilde{\boldsymbol{u}} \in B(\boldsymbol{u}^k, \delta) \cap \mathbb{R}_{\geq 0}^p$ supported on $\mathrm{supp}(\tilde{\boldsymbol{u}}) \subseteq \mathrm{supp}(\boldsymbol{u}^k) \cup \{i_k\}$, there exists a unique solution $\boldsymbol{\psi} : [0, \infty) \to \mathbb{R}^p$ of the gradient flow $\frac{d\boldsymbol{\psi}}{dt} = -\nabla_{\boldsymbol{\theta}} \mathcal{L}(\boldsymbol{\psi})$ initialized at $\boldsymbol{\psi}(0) = (\tilde{\boldsymbol{u}}, \boldsymbol{s}^{k+1} \odot \tilde{\boldsymbol{u}})$, and at times $t \geq \tau(\tilde{u}_{i_k})$,

$$\|\boldsymbol{\psi}(t) - \boldsymbol{\theta}^{k+1}\| < \epsilon \,.$$

## C.2 Rescaling time for notational convenience

For ease of notation, we rescale time

$$\boldsymbol{u}_\alpha(0) = \alpha \boldsymbol{u}(0), \quad \boldsymbol{v}_\alpha(0) = \alpha \boldsymbol{v}(0)$$

$$\frac{d\boldsymbol{u}_\alpha}{dt} = \log(1/\alpha) \boldsymbol{v}_\alpha \odot \boldsymbol{g}(\boldsymbol{u}_\alpha, \boldsymbol{v}_\alpha), \quad \frac{d\boldsymbol{v}_\alpha}{dt} = \log(1/\alpha) \boldsymbol{u}_\alpha \odot \boldsymbol{g}(\boldsymbol{u}_\alpha, \boldsymbol{v}_\alpha). \tag{14}$$

We also define

$$\boldsymbol{\theta}_\alpha(t) = (\boldsymbol{u}_\alpha(t), \boldsymbol{v}_\alpha(t)) \in \mathbb{R}^{2p} \,.$$

Because of this time-rescaling, we equivalently state Theorem 4.1 as:

**Theorem C.4** (Restatement of Theorem 4.1). *Let $K \in \mathbb{Z}_{\geq 0}$ be such that Assumptions 4.3 4.4 hold for all $k \leq K$ and Assumption 4.5 holds for all $k < K$. Then for any $k \leq K$ and time $t \in (T_k, T_{k+1})$ the following holds. There is $\alpha_0(t) > 0$ such that for all $\alpha < \alpha_0$, there exists a unique solution $\boldsymbol{\theta}_\alpha : [0, t] \to \mathbb{R}^p$ to the gradient flow (14) and*

$$\lim_{\alpha \to 0} \boldsymbol{\theta}_\alpha(t) \to \boldsymbol{\theta}^k \,,$$

*where at each stage $|\mathrm{supp}(\boldsymbol{u}^k) \setminus \mathrm{supp}(\boldsymbol{u}^{k-1})| \leq 1$.*

For shorthand, we also write

$$S_k = \mathrm{supp}(\boldsymbol{u}^k) \text{ and } S_k^c = [p] \setminus \mathrm{supp}(\boldsymbol{u}^k) \,.$$

## C.3 Simplifying problem without loss of generality

For each coordinate $i \in [p]$ we have $|u_{\alpha,i}(0)| \neq |v_{\alpha,i}(0)|$ by the non-degeneracy Assumption 4.3. So we can assume $|u_{\alpha,i}(0)| > |v_{\alpha,i}(0)|$ without loss of generality. Furthermore, we can assume the entrywise inequality

$$\boldsymbol{u}_\alpha(0) > 0$$

by otherwise training weights $\tilde{\boldsymbol{u}}_\alpha(t), \tilde{\boldsymbol{v}}_\alpha(t)$ initialized at $\tilde{\boldsymbol{u}}_\alpha(0) = \mathrm{sgn}(\boldsymbol{u}_\alpha(0))\boldsymbol{u}_\alpha(0)$ and $\tilde{\boldsymbol{v}}_\alpha(0) = \mathrm{sgn}(\boldsymbol{v}_\alpha(0))\boldsymbol{v}_\alpha(0)$, as $\tilde{\boldsymbol{u}}_\alpha(t) \odot \tilde{\boldsymbol{v}}_\alpha(t) = \boldsymbol{u}_\alpha(t) \odot \boldsymbol{v}_\alpha(t)$ at all times.

Since $u_{\alpha,i}^2(t) - v_{\alpha,i}^2(t) = u_{\alpha,i}^2(0) - v_{\alpha,i}^2(0)$ by the conservation law (5), it holds that $|u_{\alpha,i}(t)| > |v_{\alpha,i}(t)|$ throughout. So by continuity

$$\boldsymbol{u}_\alpha(t) > 0$$

throughout training.

## C.4 Tracking the sum of the weights

We define

$$\boldsymbol{w}_\alpha(t) = \boldsymbol{u}_\alpha(t) + \boldsymbol{v}_\alpha(t) \,.$$

The reason for this definition is that during training we have

$$\frac{d\boldsymbol{w}_\alpha}{dt} = \log(1/\alpha)\boldsymbol{w}_\alpha \odot \boldsymbol{g}(\boldsymbol{\theta}_\alpha) \,, \tag{15}$$

Notice that since that we have assumed $u_{\alpha,i}(0) > |v_{\alpha,i}(0)|$ for each $i \in [p]$ we have $\boldsymbol{w}_\alpha(0) > 0$ entrywise. So, by (15) for all $t > 0$,

$$\boldsymbol{w}_\alpha(t) > 0 \,.$$

It suffices to track $\boldsymbol{w}_\alpha(t)$ because we can relate the log-scale magnitude of $\boldsymbol{w}_\alpha(t)$ to the magnitudes of the corresponding coordinates in $\boldsymbol{u}_\alpha(t)$ and $\boldsymbol{v}_\alpha(t)$ – see technical Lemmas D.1 D.2 and D.3.

## C.5 Claimed invariants in proof of Theorem C.4

In order to prove Theorem C.4, we consider any gradient flow $\boldsymbol{\theta}_\alpha : [0, T^*] \to \mathbb{R}^p$ solving (14) where $T^* \in (T_K, T_{K+1})$. For now, we focus only on proving properties of this gradient flow, and defer its existence and uniqueness to Section C.8.

We show the following invariants inductively on the stage $k$. For any $\epsilon > 0$, any stage $k \leq K$, there is $\alpha_k := \alpha_k(\epsilon) > 0$ such that for all $\alpha < \alpha_k$ the following holds. There are times $\bar{t}_k := \bar{t}_k(\alpha, \epsilon)$ and $\underline{t}_{k+1} := \underline{t}_{k+1}(\alpha, \epsilon)$, such that

$$\bar{t}_k \in [T_k - \epsilon, T_k + \epsilon] \,, \tag{16}$$

$$\underline{t}_{k+1} \in \begin{cases} [T_{k+1} - \epsilon, T_{k+1} + \epsilon] \,, & \text{if } T_{k+1} < \infty \\ \{T^*\}, & \text{if } T_{k+1} = \infty \end{cases} . \tag{17}$$

and the weights approximate the greedy limit for all times $t \in [\bar{t}_k, \underline{t}_{k+1}]$

$$\|\boldsymbol{\theta}_\alpha(t) - \boldsymbol{\theta}^k\| < \epsilon \,, \tag{18}$$

and the weights at times $\bar{t}_k$ and $\underline{t}_{k+1}$ are correctly estimated by the incremental learning dynamics on the logarithmic-scale

$$\|\log_\alpha(\boldsymbol{w}_\alpha(\bar{t}_k)) - \boldsymbol{b}^k\| < \epsilon \tag{19}$$

and if $T_{k+1} < \infty$ then

$$\|\log_\alpha(\boldsymbol{w}_\alpha(\underline{t}_{k+1})) - \boldsymbol{b}^{k+1}\| < \epsilon \,. \tag{20}$$

*Base case* $k = 0$: Take $\bar{t}_0(\alpha, \epsilon) = 0$. Then statement (16) holds since $T_0 = 0$. Notice that as $\alpha \to 0$ we have that $\boldsymbol{u}_\alpha(0), \boldsymbol{v}_\alpha(0) \to \boldsymbol{0} = \boldsymbol{u}^0$, and also $\log_\alpha \boldsymbol{w}_\alpha(0) \to \boldsymbol{1} = \boldsymbol{b}^0$. So statement (19) follows if we take $\alpha_0$ small enough. In Section C.6 we show how to construct time $\underline{t}_1$ such that (18) and (20) hold.

*Inductive step*: Suppose that (16), (18), (19) and (20) hold for some iteration $k < K$. We prove them for iteration $k + 1$. In Section C.7 we construct time $\bar{t}_k$. In Section C.6 we construct time $\underline{t}_{k+1}$.

**C.6  Dynamics from time $\bar{t}_k$ to time $\underline{t}_{k+1}$ (Linear dynamics for $O(\log(1/\alpha))$ unrescaled time)**

Let $k \leq K$, and suppose that we know that for any $\bar{\epsilon}_k > 0$, there is $\bar{\alpha}_k(\bar{\epsilon}_k) > 0$ such that for all $0 < \alpha < \bar{\alpha}_k$, there is a time $\bar{t}_k = \bar{t}_k(\alpha, \bar{\epsilon}_k)$ satisfying

$$|T_k - \bar{t}_k| < \bar{\epsilon}_k$$
$$\|\boldsymbol{\theta}_\alpha(\bar{t}_k) - \boldsymbol{\theta}^k\| < \bar{\epsilon}_k$$
$$\|\log_\alpha(\boldsymbol{w}_\alpha(\bar{t}_k)) - \boldsymbol{b}^k\| < \bar{\epsilon}_k \,.$$

**C.6.1  Analysis in case where $T_{k+1} < \infty$**

Consider first the case where $T_{k+1} < \infty$. We show that, for any $\underline{\epsilon}_{k+1} > 0$, there is $\rho_{k+1}(\underline{\epsilon}_{k+1}) > 0$ such that for all $0 < \rho < \rho_{k+1}(\bar{\epsilon}_{k+1})$ there is $\underline{\alpha}_{k+1}(\rho, \underline{\epsilon}_{k+1}) > 0$ such that for all $\alpha < \underline{\alpha}_{k+1}$, there is a time $\underline{t}_{k+1} = \underline{t}_{k+1}(\alpha, \rho, \underline{\epsilon}_{k+1})$ satisfying

$$|T_{k+1} - \underline{t}_{k+1}| < \underline{\epsilon}_{k+1} \tag{21}$$
$$\|\boldsymbol{\theta}_\alpha(t) - \boldsymbol{\theta}^k\| < \underline{\epsilon}_{k+1} \text{ for all } t \in [\bar{t}_k, \underline{t}_{k+1}] \tag{22}$$
$$\|\log_\alpha(\boldsymbol{w}_\alpha(\underline{t}_{k+1})) - \boldsymbol{b}^{k+1}\| < \underline{\epsilon}_{k+1} \tag{23}$$
$$u_{\alpha, i_k}(\underline{t}_{k+1}) \in [\rho, 3\rho] \,, \tag{24}$$
$$\mathrm{sgn}(v_{\alpha, i_k}(\underline{t}_{k+1})) = s_{i_k}^{k+1} \,. \tag{25}$$

For any $\rho, \alpha$, let $\bar{\epsilon}_k = \rho\underline{\epsilon}_{k+1}/(4p)$ and choose $\bar{t}_k = \bar{t}_k(\alpha, \bar{\epsilon}_k)$. Then define

$$\underline{t}_{k+1} = \underline{t}_{k+1}(\alpha, \rho, \underline{\epsilon}_{k+1}) \tag{26}$$
$$= \inf\{t \in [\bar{t}_k, \infty) : \|\boldsymbol{u}_{\alpha, S_k^c}(t) - \boldsymbol{u}_{\alpha, S_k^c}(\bar{t}_k)\| + \|\boldsymbol{v}_{\alpha, S_k^c}(t) - \boldsymbol{v}_{\alpha, S_k^c}(\bar{t}_k)\| > 4\rho\} \,.$$

Now we show that the weights $\boldsymbol{\theta}_\alpha(t)$ cannot move much from time $\bar{t}_k$ to $\underline{t}_{k+1}$. The argument uses the local Lipschitzness of the loss $\mathcal{L}$ (from technical Lemma D.7), and the strictness of $\boldsymbol{\theta}^k$ as a stationary point (from Assumption 4.4).

**Lemma C.5** (Stability of active variables during part (A) of dynamics)**.** *There is $\rho_{k+1}$ small enough and $\underline{\alpha}_{k+1}(\rho)$ small enough depending on $\rho$,such that for all $\rho < \rho_{k+1}$ and $\alpha < \underline{\alpha}_{k+1}$ and all $t \in [\bar{t}_k, \underline{t}_{k+1})$,*

$$\|\boldsymbol{\theta}_\alpha(t) - \boldsymbol{\theta}^k\| < \rho' := \max(24\rho, 18\sqrt{\rho K_{R_k}/c_k}) \,. \tag{27}$$

*where $c_k$ is the strict-minimum constant from Assumption 4.4 and $K_{R_k}$ is the Lipschitzness constant from Lemma D.7 for the ball of radius $R_k = \|\boldsymbol{\theta}^k\| + 1$.*

*Proof.* Assume by contradiction that (27) is violated at some time $t < \underline{t}_{k+1}$. Let us choose the first such time

$$t^* = \inf\{t \in [\bar{t}_k, \underline{t}_{k+1}) : \|\boldsymbol{u}_\alpha(t^*) - \boldsymbol{u}^k\| + \|\boldsymbol{v}_\alpha(t^*) - \boldsymbol{s}^k \odot \boldsymbol{u}^k\| \geq \rho'\} \,.$$

Define $\tilde{\boldsymbol{\theta}} = (\tilde{\boldsymbol{u}}, \tilde{\boldsymbol{v}})$ by

$$\tilde{u}_i = \begin{cases} u_{\alpha, i}(t^*), & i \in S_k \\ 0, & i \notin S_k \end{cases} \quad \text{and} \quad \tilde{v}_i = \begin{cases} v_{\alpha, i}(t^*), & i \in S_k \\ 0, & i \notin S_k \end{cases} \,.$$

By the definition of $\underline{t}_{k+1}$, this satisfies

$$\|\tilde{\boldsymbol{u}} - \boldsymbol{u}_\alpha(t^*)\| = \|\boldsymbol{u}_{\alpha, S_k^c}(t^*)\| \leq 4\rho + \|\boldsymbol{u}_{\alpha, S_k^c}(\bar{t}_k)\| \leq 4\rho + \underline{\epsilon}_k < 5\rho \,,$$
$$\|\tilde{\boldsymbol{v}} - \boldsymbol{v}_\alpha(t^*)\| = \|\boldsymbol{v}_{\alpha, S_k^c}(t^*)\| \leq 4\rho + \|\boldsymbol{v}_{\alpha, S_k^c}(\bar{t}_k)\| \leq 4\rho + \underline{\epsilon}_k < 5\rho \,.$$

Also

$$\|\tilde{\boldsymbol{u}} - \boldsymbol{u}^k\| + \|\tilde{\boldsymbol{v}} - \boldsymbol{s}^k \odot \boldsymbol{u}^k\| = \|\boldsymbol{u}_{\alpha, S_k}(t^*) - \boldsymbol{z}_{S_k}^k\| + \|\boldsymbol{v}_{\alpha, S_k}(t^*) - \boldsymbol{s}_{S_k}^k \odot \boldsymbol{z}_{S_k}^k\| \geq \rho' - 10\rho \geq \rho'/2 \,.$$

Using (a) the strict minimum Assumption 4.4 with constant $c_k$, since $\|\tilde{\boldsymbol{\theta}} - \boldsymbol{\theta}^k\| \le \rho'$ and we take $\rho'$ small enough,

$$\mathcal{L}(\boldsymbol{\theta}_\alpha(t^*)) \ge \mathcal{L}(\tilde{\boldsymbol{\theta}}) - 4\rho K_{R_k} \overset{(a)}{\ge} \mathcal{L}(\boldsymbol{\theta}^k) - 4\rho K_{R_k} + \frac{c_k(\rho')^2}{16}$$

$$\ge \mathcal{L}(\boldsymbol{\theta}_\alpha(\bar{t}_k)) - (4\rho + \bar{\epsilon}_k)K_{R_k} + \frac{c_k(\rho')^2}{16} > \mathcal{L}(\boldsymbol{\theta}_\alpha(\bar{t}_k)).$$

This is a contradiction because $\mathcal{L}$ is nondecreasing along the gradient flow. $\qquad\square$

**Lemma C.6** (Log-scale approximation is correct during part (A))**.** *There are functions $\rho_{k+1}(\underline{\epsilon}_{k+1}) > 0$ and $\underline{\alpha}_{k+1}(\rho, \underline{\epsilon}_{k+1}) > 0$ such that for all $\rho < \rho_{k+1}$ and $\alpha < \underline{\alpha}_{k+1}$, and for all $t \in (\bar{t}_k, \underline{t}_{k+1})$ we have for a constant $C$ depending on $k$,*

$$\|\log_\alpha(\boldsymbol{w}_\alpha(t)) - \boldsymbol{b}^k + (t - \bar{t}_k)\boldsymbol{g}(\boldsymbol{\theta}^k)\| < \rho\underline{\epsilon}_{k+1} + C\rho'(t - \bar{t}_k). \tag{28}$$

*Furthermore, for all $i \in S_k^c$ and $t \in (\bar{t}_k, \underline{t}_{k+1})$ we have*

$$\operatorname{sgn}(g_i(\boldsymbol{\theta}_\alpha(t))) = \operatorname{sgn}(g_i(\boldsymbol{\theta}^k)). \tag{29}$$

*Proof.* By Lemma C.5 and Lemma D.7, there is a constant $C$ depending on $\boldsymbol{\theta}^k$ such that for all $t \in (\bar{t}_k, \underline{t}_{k+1})$,

$$\|\boldsymbol{g}(\boldsymbol{\theta}_\alpha(t)) - \boldsymbol{g}(\boldsymbol{\theta}^k)\| \le C\rho'.$$

For shorthand, write $\bar{\boldsymbol{g}}(\boldsymbol{\theta}^k) = \boldsymbol{g}(\boldsymbol{\theta}^k) + C\rho'\mathbf{1}$ and $\underline{\boldsymbol{g}}(\boldsymbol{\theta}^k) = \boldsymbol{g}(\boldsymbol{\theta}^k) - C\rho'\mathbf{1}$. Since $\boldsymbol{w}_\alpha(t) > 0$ entrywise as we have assumed without loss of generality (see Section C.3), we have the following entrywise inequalities

$$\underline{\boldsymbol{g}}(\boldsymbol{\theta}^k) \odot \boldsymbol{w}_\alpha(t) < \boldsymbol{g}(\boldsymbol{\theta}_\alpha(t)) \odot \boldsymbol{w}_\alpha(t) < \bar{\boldsymbol{g}}(\boldsymbol{\theta}^k) \odot \boldsymbol{w}_\alpha(t). \tag{30}$$

Since the dynamics are given by $\frac{d\boldsymbol{w}_\alpha}{dt} = \log(1/\alpha)\boldsymbol{g}(\boldsymbol{w}_\alpha) \odot \boldsymbol{w}_\alpha$,

$$\boldsymbol{w}_\alpha(\bar{t}_k)e^{(t-\bar{t}_k)\log(1/\alpha)\underline{\boldsymbol{g}}(\boldsymbol{\theta}^k)} \le \boldsymbol{w}_\alpha(t) \le \boldsymbol{w}_\alpha(\bar{t}_k)e^{(t-\bar{t}_k)\log(1/\alpha)\bar{\boldsymbol{g}}(\boldsymbol{\theta}^k)}.$$

Taking the logarithms with base $\alpha \in (0, 1)$,

$$(t - \bar{t}_k)\underline{\boldsymbol{g}}(\boldsymbol{u}^k) \le \log_\alpha(\boldsymbol{w}_\alpha(\bar{t}_k)) - \log_\alpha(\boldsymbol{w}_\alpha(t)) \le (t - \bar{t}_k)\bar{\boldsymbol{g}}(\boldsymbol{u}^k).$$

The bound (28) follows since $\|\log_\alpha(\boldsymbol{w}_\alpha(\bar{t}_k)) - \boldsymbol{b}^k\| < \bar{\epsilon}_k < \rho\underline{\epsilon}_{k+1}$.

Finally, the claim (29) follows from (30) since $\operatorname{sgn}(\bar{\boldsymbol{g}}(\boldsymbol{\theta}^k)) = \operatorname{sgn}(\underline{\boldsymbol{g}}(\boldsymbol{\theta}^k)) = \operatorname{sgn}(\boldsymbol{g}(\boldsymbol{\theta}^k))$ if we take $\rho$ small enough. $\qquad\square$

First, we show that the weights must move significantly by time roughly $T_{k+1}$. This is because of the contribution of coordinate $i_k$.

**Lemma C.7** ($\underline{t}_{k+1}$ is not much larger than $T_{k+1}$)**.** *Suppose that $T_{k+1} < \infty$. Then there are $\rho_{k+1}(\underline{\epsilon}_{k+1}) > 0$ and $\underline{\alpha}_{k+1}(\rho, \underline{\epsilon}_{k+1}) > 0$ such that for all $\rho < \rho_{k+1}$ and $\alpha < \underline{\alpha}_{k+1}$, the following holds.*

$$\underline{t}_{k+1} < T_{k+1} + \underline{\epsilon}_{k+1}.$$

*Proof.* Assume by contradiction that $\underline{t}_{k+1} < T_{k+1} + \underline{\epsilon}_{k+1}$. For all times $t \in [\bar{t}_k, \min(\underline{t}_{k+1}, T_{k+1} + \underline{\epsilon}_{k+1})]$, by Lemma C.6,

$$|\log_\alpha(w_{\alpha,i_k}(t)) - b_{i_k}^t + (t - \bar{t}_k)g_{i_k}(\boldsymbol{\theta}^k)| < O(\sqrt{\rho}).$$

Since we know $|\Delta_k(i_k) - (T_{k+1} - \bar{t}_k)| < \bar{\epsilon}_k$ and $b_i^k - \Delta_k(i_k)g_{i_k}(\boldsymbol{\theta}^k) \in \{0, 2\}$, it follows that

$$\log_\alpha(w_{\alpha,i_k}(T_{k+1} + \underline{\epsilon}_{k+1})) \notin (-|g_{i_k}(\boldsymbol{\theta}^k)|(\underline{\epsilon}_{k+1} - \bar{\epsilon}_{k+1}), 2 + |g_{i_k}(\boldsymbol{\theta}^k)|(\underline{\epsilon}_{k+1} - \bar{\epsilon}_{k+1})) + O(\sqrt{\rho}).$$

By taking $\rho$ small enough, we see that $|g_{i_k}(\boldsymbol{\theta}^k)|(\underline{\epsilon}_{k+1} - \bar{\epsilon}_{k+1}) + O(\sqrt{\rho}) > \delta > 0$ for some $\delta > 0$ that is independent of $\alpha$, so

$$\log_\alpha(w_{\alpha,i_k}(T_{k+1} + \underline{\epsilon}_{k+1})) \notin (-\delta, 2 + \delta).$$

So $|u_{\alpha,i_k}(T_{k+1} + \underline{\epsilon}_{k+1})| > 1$ by Lemma D.2. But by the construction of $\underline{t}_{k+1}$ this means that $\underline{t}_{k+1} < T_{k+1} + \underline{\epsilon}_{k+1}$. $\qquad\square$

Next, we show that until time $\underline{t}_{k+1}$, none of the coordinates in $S_k^c$ move significantly, with the possible exception of coordinate $i_k$.

**Lemma C.8** (No coordinates in $S_k^c \setminus \{i_k\}$ move significantly during part (A)). *Suppose $T_{k+1} < \infty$. Then there are $\rho_{k+1}(\underline{\epsilon}_{k+1}) > 0$ and $\underline{\alpha}_{k+1}(\rho, \underline{\epsilon}_{k+1}) > 0$ such that for all $\rho < \rho_{k+1}$ and $\alpha < \underline{\alpha}_{k+1}$, the following holds. There is a constant $c > 0$ depending on $k$ such that for all $i \in S_k^c \setminus \{i_k\}$ and $t \in [\bar{t}_k, \underline{t}_{k+1}]$,*

$$|u_{\alpha,i}(t) - u_{\alpha,i}(\bar{t}_k)|, |v_{\alpha,i}(t) - v_{\alpha,i}(\bar{t}_k)| < \alpha^c + \bar{\epsilon}_k \,.$$

*Proof.* The previous lemma combined with the inductive hypothesis gives

$$\underline{t}_{k+1} - \bar{t}_k < \Delta_k(i_k) + 2\underline{\epsilon}_{k+1} \setminus \{i_k\}.$$

We analyze the movement of each coordinate $i \in S_k^c \setminus \{i_k\}$ by breaking into two cases:

- Coordinate $i \neq i_k$ such that $b_i^k \in (0,2)$. By Assumption 4.3, there is a unique winning coordinate so $b_i^k - \tau g_i(\boldsymbol{\theta}^k) \in (c, 2-c)$ for some constant $c > 0$ for all $\tau \in [0, \underline{t}_{k+1} - \bar{t}_k] \subseteq [0, \Delta_k(i_k) + 2\underline{\epsilon}_{k+1}]$. By Lemma C.6, $\log_\alpha(w_{\alpha,i}(t)) \in (-c/2, 2 - c/2)$ for all times $t \in [\bar{t}_k, \underline{t}_{k+1}]$. So by Lemma D.1, $|u_{\alpha,i}(t)|, |v_{\alpha,i}(t)| \leq \alpha^{c/4}$.

- Coordinate $i \neq i_k$ such that $b_i^k = 0$. By Lemma D.4, we must be in the corner case where $i \in S_{k-1} \cap S_k^c$ (i.e., the coordinate was active in the previous stage but was dropped from the support in this stage).

  By Lemma D.4, since $b_i^k = 0$ we have $g_i(\boldsymbol{\theta}^k) < 0$. By Lemma C.6, this means $\text{sgn}(g_i(\boldsymbol{\theta}_\alpha(t))) = \text{sgn}(g_i(\boldsymbol{\theta}^k)) < 0$ for all $t \in (\bar{t}_k, \underline{t}_{k+1})$.

  We break the analysis into two parts. Since $b_i^k = 0$, the sign is $s_i^k = +1$. The inductive hypothesis $\|\boldsymbol{\theta}_\alpha(\bar{t}_k) - \boldsymbol{\theta}^k\| < \bar{\epsilon}_k$ implies that $|u_{\alpha,i}(\bar{t}_k) - z_i^k| < \bar{\epsilon}_k$ and $|v_{\alpha,i}(\bar{t}_k) - z_i^k| < \bar{\epsilon}_k$. For small enough $\bar{\epsilon}_k$ this means that $\text{sgn}(u_{\alpha,i}(\bar{t}_k)) = \text{sgn}(v_{\alpha,i}(\bar{t}_k)) = +1$. Now let $t^* = \min(\underline{t}_{k+1}, \inf\{t > \bar{t}_k : v_{\alpha,i}(t) = 0\})$. Since $u_{\alpha,i}(t) > v_{\alpha,i}(t)$ without loss of generality (see Section C.3), we have $\text{sgn}(u_{\alpha,i}(t)) = \text{sgn}(v_{\alpha,i}(t)) = +1$ for all $t \in [\bar{t}_k, t^*]$. So $\frac{du_{\alpha,i}(t)}{dt}, \frac{dv_{\alpha,i}(t)}{dt} < 0$ for all $t \in [\bar{t}_k, t^*]$. So, for any $t \in [\bar{t}_k, t^*]$,

  $$|u_{\alpha,i}(t) - u_{\alpha,i}(\bar{t}_k)|, |v_{\alpha,i}(t) - v_{\alpha,i}(\bar{t}_k)| < \bar{\epsilon}_k$$

  Also, since $\log_\alpha(w_{\alpha,i}(t^*)) \approx 1$, by Lemma C.6 we have $t^* > c > 0$ for some constant $c$ independent of $\alpha$. So for all $t \in [t^*, \underline{t}_{k+1}]$ we have $b_i^k - \tau g_i(\boldsymbol{\theta}^k) \in (c, 2-c)$ for some constant $c > 0$. So $|u_{\alpha,i}(t)|, |v_{\alpha,i}(t)| \leq \alpha^{c/4}$ for all $t \in [t^*, \underline{t}_{k+1}]$. The conclusion follows by triangle inequality.

- Coordinate $i \neq i_k$ such that $b_i^k = 2$. The analysis is analogous to the case $b_i^k = 0$, except that we have $s_i^k = -1$ instead and $g_i(\boldsymbol{\theta}^k) > 0$ by Lemma D.4.

$\square$

Finally, we use this conclude that $\underline{t}_{k+1} \approx T_{k+1}$ and that the weights at coordinate $i_k$ are the only weights that change significantly, and by an amount approximately $\rho$.

**Lemma C.9** (Coordinate $i_k$ wins the part (A) race at time $\underline{t}_{k+1} \approx T_{k+1}$). *Suppose that $T_{k+1} < \infty$. Then there are $\rho_{k+1}(\underline{\epsilon}_{k+1}) > 0$ and $\underline{\alpha}_{k+1}(\rho, \underline{\epsilon}_{k+1}) > 0$ such that for all $\rho < \rho_{k+1}$ and $\alpha < \underline{\alpha}_{k+1}$, the following holds.*

$$|\underline{t}_{k+1} - T_{k+1}| < \underline{\epsilon}_{k+1}\,,$$

$$u_{\alpha,i_k}(\underline{t}_{k+1}) \in [\rho, 3\rho]\,,$$

$$\text{sgn}(v_{\alpha,i_k}(\underline{t}_{k+1})) = s_{i_k}^{k+1}\,.$$

*Proof.* Let us analyze the case that $b_{i_k}^k \in (0,2)$. Notice that $b_{i_k}^{k+1} = b_{i_k}^k - \Delta_k(i_k)g_{i_k}(\boldsymbol{\theta}^k) \in \{0,2\}$ and that if $b_i^{k+1} = 0$ then $g_{i_k}(\boldsymbol{\theta}^k) > 0$ and if it is 2 then $b_{i_k}^{k+1} = g_{i_k}(\boldsymbol{\theta}^k) < 0$. So by Lemma C.6, for all times $t \in [\bar{t}_k, \min(\underline{t}_{k+1}, T_{k+1} - \underline{\epsilon}_{k+1})]$, we have $w_{\alpha,i_k}(t) \in (c, 2-c)$ for some $c > 0$. So for small enough $\alpha$ by Lemma D.1, $|u_{\alpha,i_k}(t)|, |v_{\alpha,i_k}(t)| \le \alpha^{c/2}$. Combining this with Lemma C.8, we see that for $t \in [\bar{t}_k, \min(\underline{t}_{k+1}, T_{k+1} - \underline{\epsilon}_{k+1})]$ we have

$$\|\boldsymbol{u}_\alpha(t) - \boldsymbol{u}_\alpha(\bar{t}_k)\| + \|\boldsymbol{v}_\alpha(t) - \boldsymbol{v}_\alpha(\bar{t}_k)\| < 2(\alpha^c + \bar{\epsilon}_k)p < \rho\,,$$

for small enough $\alpha$. So by definition of $\underline{t}_{k+1}$ we must have $\underline{t}_{k+1} > T_{k+1} - \underline{\epsilon}_{k+1}$. Combined with Lemma C.7, we conclude that $|T_{k+1} - \underline{t}_{k+1}| < \underline{\epsilon}_{k+1}$, which is the first claim of the lemma. Furthermore, by Lemma C.8,

$$\sum_{i \in S_k^c \setminus \{i_k\}} |u_{\alpha,i}(\underline{t}_{k+1}) - u_{\alpha,i}(\bar{t}_k)| + |v_{\alpha,i}(\underline{t}_{k+1}) - v_{\alpha,i}(\bar{t}_k)| \le 2p(\alpha^c + \bar{\epsilon}_k)) < \rho/2,$$

so by definition of $\underline{t}_{k+1}$ and triangle inequality we have $|u_{\alpha,i_k}(\underline{t}_{k+1})| + |v_{\alpha,i_k}(\underline{t}_{k+1})| \ge 4\rho - \rho/2 = 7\rho/2$. Also, since $u_{\alpha,i_k}^2(\underline{t}_{k+1}) - v_{\alpha,i_k}^2(\underline{t}_{k+1}) = \Theta(\alpha^2)$ we have $u_{\alpha,i_k}(\underline{t}_{k+1}) \in [\rho, 3\rho]$. Finally, if $b_{i_k}^{k+1} = 2$, then $s_{i_k}^{k+1} = -1$ and $\log_\alpha(w_{\alpha,i_k}(\underline{t}_{k+1})) > 1.5$ so $\text{sgn}(v_{\alpha,i_k}(t)) < 0$ by Lemma D.3; analogously, if $b_{i_k}^{k+1} = 0$, we have $s_{i_k}^{k+1} = 1$ and $\log_\alpha(w_{\alpha,i_k}(\underline{t}_{k+1}) < 0.5$ so $\text{sgn}(v_{\alpha,i_k}(\underline{t}_{k+1}) > 0$.

The case $b_{i_k}^k \in \{0,2\}$ can be proved similarly to the analysis in Lemma C.8, where one shows that during the first period of time the magnitudes of $|u_{i_k}(t)|$ and $|v_{i_k}(t)|$ decrease, until the sign of $v_{i_k}$ flips and they once again increase.

$\square$

We have shown the claims (21), (22), (23) (24), and (25) for the time $\underline{t}_{k+1}$. In fact, if we let $\underline{t}'_{k+1} \in [\bar{t}_k, \infty)$ be the first time $t$ such that $u_{\alpha,i_k}(t) = \rho$ we still have (21), (22), (23) and (25) by the same analysis as above, and (24) can be replaced with the slightly more convenient

$$u_{\alpha,i_k}(\underline{t}'_{k+1}) = \rho\,.$$

### C.6.2 Analysis in case where $T_{k+1} = \infty$

In this case that $T_{k+1}$, we just have to show that the weights remain close to $\boldsymbol{\theta}^k$. We show that for any $\underline{\epsilon}_{k+1} > 0$, there is $\underline{\alpha}_{k+1}(\underline{\epsilon}_{k+1}) > 0$ such that for all $\alpha < \underline{\alpha}_{k+1}$ and times $t \in [T_k + \underline{\epsilon}_{k+1}, T^*]$,

$$\|\boldsymbol{\theta}_\alpha(t) - \boldsymbol{\theta}^k\| < \underline{\epsilon}_{k+1}.$$

We can use Lemmas C.5 and C.6, which were developed for the case of $T_{k+1} < \infty$, but still hold for $T_{k+1} = \infty$. Lemma C.5 guarantees that the weights do not move much until time $\underline{t}_{k+1}$, and so we only need to show that $\underline{t}_{k+1} \ge T^*$ when we take $\rho$ small enough. For this, observe that $g_i(\boldsymbol{\theta}^k) = 0$ for all $i \notin S_k$, because otherwise $T_{k+1} < \infty$. Therefore Lemma C.6 guarantees that until time $\min(T_*, \underline{t}_{k+1})$ all weights are close to the original on the logarithmic scale. Namely,

$$\| \log_\alpha(\boldsymbol{w}_\alpha(t)) - \boldsymbol{b}^k\| < \rho\underline{\epsilon}_{k+1} + C\rho'(T^* - \bar{t}_k)$$

Furthermore, by the non-degeneracy Assumption 4.3 we know that $b_i^k \in (0,2)$ for all $i \notin S_k$ by Lemma D.4. So if we take $\rho$ small enough and $\underline{\alpha}_{k+1}$ small enough, we must have that $\underline{t}_{k+1} \ge T^*$.

### C.7 Dynamics from time $\underline{t}_k$ to time $\bar{t}_k$ (Nonlinear evolution for $O(1)$ unrescaled time)

Suppose that we know for some $k \le K$ that for any $\underline{\epsilon}_k > 0$, there is $\rho_k(\underline{\epsilon}_k) > 0$ such that for all $\rho < \rho_k$ there is $\underline{\alpha}_k(\rho, \underline{\epsilon}_k) > 0$ such that for all $\alpha < \underline{\alpha}_k$, there is a time $\underline{t}_k = \underline{t}_k(\alpha, \rho, \underline{\epsilon}_k)$ satisfying

$$|T_k - \underline{t}_k| < \underline{\epsilon}_k \tag{31}$$

$$\|\boldsymbol{\theta}_\alpha(\underline{t}_k) - \boldsymbol{\theta}^{k-1}\| < \underline{\epsilon}_k \tag{32}$$

$$\| \log_\alpha(\boldsymbol{w}_\alpha(\underline{t}_k)) - \boldsymbol{b}^k\| < \underline{\epsilon}_k \tag{33}$$

$$u_{\alpha,i_{k-1}}(\underline{t}_k) = \rho\,, \tag{34}$$

$$\text{sgn}(v_{\alpha,i_{k-1}}(\underline{t}_k)) = s_{i_{k-1}}^k\,. \tag{35}$$

Now we will show that for any $\bar{\epsilon}_k > 0$, there is $\bar{\alpha}_k = \bar{\alpha}_k(\bar{\epsilon}_k) > 0$ such that for all $0 < \alpha < \bar{\alpha}_k$, there is a time $\bar{t}_k = \bar{t}_k(\alpha, \bar{\epsilon}_k)$ satisfying

$$|T_k - \bar{t}_k| < \bar{\epsilon}_k \tag{36}$$

$$\|\boldsymbol{\theta}_\alpha(\bar{t}_k) - \boldsymbol{\theta}^k\| < \bar{\epsilon}_k \tag{37}$$

$$\|\log_\alpha(\boldsymbol{w}_\alpha(\bar{t}_k)) - \boldsymbol{b}^k\| < \bar{\epsilon}_k \tag{38}$$

We give the construction for $\bar{t}_k$. For any desired accuracy $\bar{\epsilon}_k > 0$ in this stage, we will construct an accuracy $\underline{\epsilon}_k = \underline{\epsilon}_k(\bar{\epsilon}_k) = \bar{\epsilon}_k/3 > 0$. We will also construct a $\rho = \rho(\underline{\epsilon}_k) > 0$ which is sufficiently small, and we will construct an cutoff for $\alpha$ equal to $\bar{\alpha}_k = \bar{\alpha}_{k+1}(\bar{\epsilon}_k) > 0$ which satisfies $\bar{\alpha}_k < \underline{\alpha}_k(\rho, \underline{\epsilon}_k)$. The values for these parameters $\underline{\epsilon}_k$ and $\rho$ and $\bar{\alpha}_k$ will be chosen in the following lemma, and will depend only on $\bar{\epsilon}_k$.

**Lemma C.10** (New local minimum reached in time $O(1/\log(1/\alpha))$). *For any $\bar{\epsilon}_k > 0$, we can choose $\bar{\alpha}_k = \bar{\alpha}_k(\bar{\epsilon}_k) > 0$ small enough so that, for any $0 < \alpha < \bar{\alpha}_k$, there is $\bar{t}_k = \bar{t}_k(\alpha, \bar{\epsilon}_k)$ for which conditions (36) to (38) hold.*

*Furthermore, there is a constant $C''$ independent of $\alpha$ such that $|\boldsymbol{\theta}_\alpha(t)|/|\boldsymbol{\theta}_\alpha(\underline{t}_k)| \in [1/C'', C'']^{2p}$ at all times $t \in [\underline{t}_k, \bar{t}_k]$.*

*Proof.* Let $\underline{t}_k = \underline{t}_k(\alpha, \rho, \underline{\epsilon}_k)$ be given by the induction. Let us compare the dynamics starting at $\boldsymbol{\theta}_\alpha(\underline{t}_k)$ with the dynamics starting at $\tilde{\boldsymbol{\theta}}(\underline{t}_k) = (\tilde{\boldsymbol{u}}(\underline{t}_k), \tilde{\boldsymbol{v}}(\underline{t}_k))$ which is given by

$$\tilde{u}_i(\underline{t}_k) = \begin{cases} u_{\alpha,i}(\underline{t}_k), & i \in S_{k-1} \cup \{i_{k-1}\} \\ 0, & \text{otherwise} \end{cases} \quad \text{and} \quad \tilde{v}_i(\underline{t}_k) = \begin{cases} v_{\alpha,i}(\underline{t}_k), & i \in S_{k-1} \cup \{i_{k-1}\} \\ 0, & \text{otherwise} \end{cases}$$

and run with

$$\frac{d\tilde{\boldsymbol{\theta}}}{dt} = -\log(1/\alpha)\nabla_{\boldsymbol{w}}\mathcal{L}(\tilde{\boldsymbol{\theta}}) \ .$$

By Assumption 4.5 we know there exists a unique solution $\tilde{\boldsymbol{\theta}} : [\underline{t}_k, \infty) \to \mathbb{R}^p$ as long as we take $\underline{\epsilon}_k$ small enough because $\text{supp}(\tilde{\boldsymbol{\theta}}(\underline{t}_k)) = S_{k-1} \cup \{i_{k-1}\}$ and $\|\tilde{\boldsymbol{\theta}}_i(\underline{t}_k) - \boldsymbol{\theta}^{k-1}\| < \underline{\epsilon}_k$. Furthermore, by Assumption 4.5 if we take $\underline{\epsilon}_k$ small enough there must be a time $\tau := \tau(\bar{\epsilon}_k, \rho) < \infty$ such that

$$\|\tilde{\boldsymbol{\theta}}(t) - \boldsymbol{\theta}^k\| < \bar{\epsilon}_k/2 \text{ for } t \geq \underline{t}_k + \tau/\log(1/\alpha) \tag{39}$$

Define

$$\bar{t}_k = \underline{t}_k + \tau/\log(1/\alpha).$$

So for $\alpha$ small enough, $|T_k - \bar{t}_k| < 2\underline{\epsilon}_k < \bar{\epsilon}_k$, proving (36).

We now compare $\boldsymbol{\theta}_\alpha(\bar{t}_k)$ with $\tilde{\boldsymbol{\theta}}(\bar{t}_k)$, and show that if we take $\alpha$ small enough, then the dynamics of $\tilde{\boldsymbol{\theta}}$ closely match the dynamics of $\boldsymbol{\theta}_\alpha(t)$ for times $\underline{t}_k + O(1/\log(1/\alpha))$. The argument uses Gronwall's inequality. Let $t^* = \inf\{t > \underline{t}_k : \|\tilde{\boldsymbol{\theta}}(t^*) - \boldsymbol{\theta}_\alpha(t)\| > 1/3\}$. For times $t \in [\underline{t}_k, t^*)$ by Lemma D.7 we have

$$\|\frac{d}{dt}\tilde{\boldsymbol{\theta}}(t) - \frac{d}{dt}\boldsymbol{\theta}_\alpha(t)\| = \log(1/\alpha)\|\nabla_{\boldsymbol{\theta}}\mathcal{L}(\tilde{\boldsymbol{\theta}}(t)) - \nabla_{\boldsymbol{\theta}}\mathcal{L}(\boldsymbol{\theta}_\alpha(t))\| \leq K_{\tilde{\boldsymbol{\theta}}(t)}\log(1/\alpha)\|\tilde{\boldsymbol{\theta}}(t) - \boldsymbol{\theta}_\alpha(t)\|,$$

where $K_{\tilde{\boldsymbol{\theta}}(t)}$ is the smoothness constant from Lemma D.7. Note that since $\|\tilde{\boldsymbol{\theta}}(t)\| < \infty$ for large enough $t$ by (39), the trajectory of $\tilde{\boldsymbol{\theta}}$ must lie in a compact set. Therefore, there must be a finite set of times $s_1, \ldots, s_m \in [\underline{t}_k, t^*)$ such that $\cup_{t \in [\underline{t}_k, t^*)} B(\tilde{\boldsymbol{\theta}}(t), 1/2) \subseteq \cup_{i=1}^m B(\tilde{\boldsymbol{\theta}}(s_i), 3/4)$. So letting $C = \max_{i=1}^m K_{\tilde{\boldsymbol{\theta}}(s_i)} < \infty$ for all times $t \in [\underline{t}_k, t^*)$ we have

$$\frac{d}{dt}\|\tilde{\boldsymbol{\theta}}(t) - \boldsymbol{\theta}_\alpha(t)\| \leq C\log(1/\alpha)\|\tilde{\boldsymbol{\theta}}(t) - \boldsymbol{\theta}_\alpha(t)\| \ .$$

By Gronwall's inequality, for all times $t \in [\underline{t}_k, t^*)$,

$$\|\tilde{\boldsymbol{\theta}}(t) - \boldsymbol{\theta}_\alpha(t)\| \leq \|\tilde{\boldsymbol{\theta}}(\underline{t}_k) - \boldsymbol{\theta}_\alpha(\underline{t}_k)\| \exp(C\log(1/\alpha)(t - \underline{t}_k)) \ .$$

We know from Lemma C.8 that there is a constant $c > 0$ such that for any small enough $0 < \alpha < \underline{\alpha}_k$, such that

$$\|\tilde{\boldsymbol{\theta}}(\underline{t}_k) - \boldsymbol{\theta}_\alpha(\underline{t}_k)\| < \alpha^c$$

If we take $\alpha$ small enough that $\alpha^c \exp(C\tau) < \bar{\epsilon}_k/2 < 1/3$, we must have $t^* > \underline{t}_k + \tau/\log(1/\alpha)$ and so we prove (37)

$$\|\boldsymbol{\theta}^k - \boldsymbol{\theta}_\alpha(\bar{t}_k)\| \leq \bar{\epsilon}_k/2 + \|\tilde{\boldsymbol{\theta}}(\bar{t}_k) - \boldsymbol{\theta}_\alpha(\bar{t}_k)\| < \bar{\epsilon}_k .$$

It remains to show that (38) is satisfied. Since $\|\tilde{\boldsymbol{\theta}}(t) - \boldsymbol{\theta}_\alpha(t)\| < 1/3$ for all $t \in [\underline{t}_k, \bar{t}_k]$, it holds that the trajectory of $\boldsymbol{\theta}_\alpha(t)$ lies in a compact set. So by Lemma D.7 we have $\|\boldsymbol{g}(\boldsymbol{\theta}_\alpha(t))\| < C'$ for some constant $C'$ at all times $t \in [\underline{t}_k, \bar{t}_k]$. Since $\frac{1}{\log(1/\alpha)}|\frac{dw_{\alpha,i}}{dt}| = |w_{\alpha,i}(t)||g_i(\boldsymbol{w}_\alpha(t))| < C'|w_{\alpha,i}(t)|$, we must have $|w_{\alpha,i}(t)|/|w_{\alpha,i}(\underline{t}_k)| \in [1/C'', C'']$ for some constant $C''$ independent of $\alpha$ and all $t \in [\underline{t}_k, \bar{t}_k]$. Therefore, (38) follows from (33). A similar argument shows that $|\boldsymbol{\theta}_\alpha(t)/\boldsymbol{\theta}_\alpha(\underline{t}_k)| \in [1/C''', C''']^{2p}$.

$\square$

## C.8 Concluding the proof of Theorem C.4

We have shown that Theorem 4.1 is true for solutions $\boldsymbol{\theta}_\alpha : [0, T^*] \to \mathbb{R}^{2p}$ to the gradient flow, where $T_* \in (T_K, T_{K+1})$. To establish Theorem C.4 it remains only to show that for any $T_* \in (T_K, T_{K+1})$ and small enough $\alpha$ such a solution to the gradient flow exists and is unique. To see this, note that in the inductive proof of the invariants we construct a sequence of times $0 = \bar{t}_0 \leq \underline{t}_1 \leq \bar{t}_1 \leq \cdots \leq \bar{t}_K \leq \underline{t}_{K+1} > T_*$, where we guarantee that any gradient flow solution $\boldsymbol{\theta}_\alpha : [0, \underline{t}_{k+1}] \to \mathbb{R}^p$ satisfies $\boldsymbol{\theta}_\alpha \in \cup_{k \in \{0,\dots,K\}} B(\boldsymbol{\theta}^k, 1)$ for all $t \in \cup_{k \in \{0,\dots,K\}}[\bar{t}_k, \underline{t}_{k+1}]$. And also for $t \in \cup_{k \in \{0,\dots,K-1\}}[\underline{t}_k, \bar{t}_{k+1}]$, we have $\boldsymbol{\theta}_\alpha(t) \in B(0, C_k''\boldsymbol{\theta}^k)$ for some constant $C_k''$ independent of $\alpha$ by Lemma C.10. So $\boldsymbol{\theta}_\alpha(t) \in B(0, C_K)$ for some constant $C_K$ at all times $t \in [0, T^*]$. By Lemma D.7, the loss gradient $\nabla_{\boldsymbol{\theta}}\mathcal{L}(\boldsymbol{\theta}) = (\boldsymbol{v} \odot \boldsymbol{g}(\boldsymbol{\theta}), \boldsymbol{u} \odot \boldsymbol{g}(\boldsymbol{\theta}))$ is Lipschitz-continuous on the compact set $B(0, C_K)$. So $\boldsymbol{\theta}_\alpha : [0, T^*] \to \mathbb{R}^p$ exists and is unique by the Cauchy-Lipschitz theorem.

$\square$

## D Technical lemmas

### D.1 Relating the sum of the weights to the original weights using the conservation law

**Lemma D.1.** *If for some constant $0 < c < 1$ we have $\log_\alpha(w_{\alpha,i}(t)) \in (c, 2 - c)$, then for small enough $\alpha$*

$$\max(|u_{\alpha,i}(t)|, |v_{\alpha,i}(t)|) \leq \alpha^{c/2} .$$

*Proof.* Let $\tilde{\boldsymbol{w}}_\alpha(t) = \boldsymbol{u}_\alpha(t) - \boldsymbol{v}_\alpha(t)$. By the conservation law (5), $w_{\alpha,i}(t)\tilde{w}_{\alpha,i}(t) = w_{\alpha,i}(0)\tilde{w}_{\alpha,i}(0) = u_{\alpha,i}(0)^2 - v_{\alpha,i}(0)^2$. By the non-degeneracy of initialization (Assumption 4.3), the right-hand-side is $\Theta(\alpha^2)$. So if $\log_\alpha(w_{\alpha,i}(t)) \in (c, 2 - c)$ then for small enough $\alpha$, we have $\log_\alpha(|\tilde{w}_{\alpha,i}(t)|) \in (3c/4, 2 - 3c/4)$. So $|u_{\alpha,i}(t)| \leq |w_{\alpha,i}(t) + \tilde{w}_{\alpha,i}(t)| \leq \alpha^{c/2}$ and $|v_{\alpha,i}(t)| \leq |w_{\alpha,i}(t) - \tilde{w}_{\alpha,i}(t)| \leq \alpha^{c/2}$. $\square$

**Lemma D.2.** *If for some constant $0 < c$ we have $\log_\alpha(w_{\alpha,i}(t)) \notin (-c, 2 + c)$, then for small enough $\alpha$,*

$$|u_{\alpha,i}(t)| > 1 .$$

*Proof.* Define $\tilde{\boldsymbol{w}}_\alpha = \boldsymbol{u}_\alpha - \boldsymbol{v}_\alpha$ as in the proof of Lemma D.1. If $\log_\alpha(w_{\alpha,i}(t)) < -c$ then $\log_\alpha(|\tilde{w}_{\alpha,i}(t)|) > 2 - c/2$ for small enough $\alpha$, so $u_i(t) > \alpha^{-c} - \alpha^{2-c/2} > 1$. Similarly, if $\log_\alpha(w_{\alpha,i}(t)) > 2 + c$ then $\log_\alpha(|\tilde{w}_{\alpha,i}(t)|) < -c/2$ so $|u_i(\alpha)| > \alpha^{-c/2} - \alpha^{2+c} > 1$. $\square$

**Lemma D.3.** *If for some constant $c > 0$, there is small enough $\alpha$ such that if we have $\log_\alpha(w_{\alpha,i}(t)) > 1 + c$ then $\mathrm{sgn}(v_{\alpha,i}(t)) < 0$. Otherwise, if $\log_\alpha(w_{\alpha,i}(t)) < 1 - c$ then $\mathrm{sgn}(v_{\alpha,i}(t)) > 0$.*

*Proof.* Follows from $\boldsymbol{v}_\alpha = \frac{1}{2}(\boldsymbol{w}_\alpha - \tilde{\boldsymbol{w}}_\alpha)$. Recall that $\boldsymbol{w}_\alpha(t) > 0$ and notice that $\tilde{\boldsymbol{w}}_\alpha(t) > 0$. In the first case, $w_{\alpha,i}(t) < \alpha^{1+c}$ and $\tilde{w}_{\alpha,i}(t) > \alpha^{1-c/2}$. In the latter case $w_{\alpha,i}(t) > \alpha^{1-c}$ and $\tilde{w}_{\alpha,i}(t) < \alpha^{1+c/2}$. $\qquad\square$

## D.2  Sign of gradients on coordinates that leave support

**Lemma D.4.** *For any $k \geq 1$ and $i \in S_k^c$, if $b_i^k \in \{0,2\}$ then we must have $i \in \operatorname{supp}(\boldsymbol{u}^{k-1}) \setminus \operatorname{supp}(\boldsymbol{u}^k)$, and we must have $g_i(\boldsymbol{u}^k) < 0$ if $b_i^k = 0$ and $g_i(\boldsymbol{\theta}^k) > 0$ if $b_i^k = 2$. In particular, $\Delta_k(i_k) > 0$ for all $k$.*

*Proof.* This is by induction on $k$ and using the non-degeneracy Assumption 4.3. $\qquad\square$

## D.3  Local lipschitzness and smoothness

We provide several technical lemmas on the local Lipschitzness and smoothness of $\ell$, $h$, and $\boldsymbol{g}$.

**Lemma D.5.** *The function $\ell(\boldsymbol{y}, \cdot)$ is locally Lipschitz and smooth in its second argument: for any $R > 0$, there exists $K_R$ such that for any $\boldsymbol{\zeta}, \boldsymbol{\zeta}' \in B(0, R)$*

$$|\ell(\boldsymbol{y}, \boldsymbol{\zeta}) - \ell(\boldsymbol{y}, \boldsymbol{\zeta}')| \leq K_R \|\boldsymbol{\zeta} - \boldsymbol{\zeta}'\|$$
$$\|D\ell(\boldsymbol{y}, \boldsymbol{\zeta}) - D\ell(\boldsymbol{y}, \boldsymbol{\zeta}')\| \leq K_R \|\boldsymbol{\zeta} - \boldsymbol{\zeta}'\|,$$

*almost surely over $\boldsymbol{y}$. Here $D\ell(\boldsymbol{y}, \cdot)^\top \in \mathbb{R}^{d_{out}}$ is the derivative in the second argument.*

*Proof.* Since $\ell$ is continuously twice-differentiable, for each $\boldsymbol{y} \in \mathbb{R}^{d_y}, \boldsymbol{\zeta} \in \mathbb{R}^{d_{out}}$ there is $K_{\boldsymbol{y},\boldsymbol{\zeta}} < \infty$ such that for all $\boldsymbol{y} \in B(\boldsymbol{y}, 1/K_{\boldsymbol{y},\boldsymbol{\zeta}})$ and $\boldsymbol{\zeta}' \in B(\boldsymbol{\zeta}, 1/K_{\boldsymbol{y},\boldsymbol{\zeta}})$ we have

$$\|D\ell(\boldsymbol{y}', \boldsymbol{\zeta}')\| \leq K_{\boldsymbol{y},\boldsymbol{\zeta}} \quad \text{and} \quad \|D^2\ell(\boldsymbol{y}', \boldsymbol{\zeta}')\| \leq K_{\boldsymbol{y},\boldsymbol{\zeta}},$$

where $D\ell$ and $D^2\ell$ denote the first and second derivative in the second argument. So for all such $\boldsymbol{y}' \in B(\boldsymbol{y}, 1/K_{\boldsymbol{y},\boldsymbol{\zeta}})$ and $\boldsymbol{\zeta}', \boldsymbol{\zeta}'' \in B(\boldsymbol{\zeta}, 1/K_{\boldsymbol{y},\boldsymbol{\zeta}})$ we have

$$|\ell(\boldsymbol{y}', \boldsymbol{\zeta}') - \ell(\boldsymbol{y}', \boldsymbol{\zeta}'')| \leq K_{\boldsymbol{y},\boldsymbol{\zeta}} \|\boldsymbol{\zeta}' - \boldsymbol{\zeta}''\| \quad \text{and} \quad |D\ell(\boldsymbol{y}', \boldsymbol{\zeta}') - D\ell(\boldsymbol{y}', \boldsymbol{\zeta}'')| \leq K_{\boldsymbol{y},\boldsymbol{\zeta}} \|\boldsymbol{\zeta}' - \boldsymbol{\zeta}''\|.$$

Cover the set $\{(\boldsymbol{y}, \boldsymbol{\zeta}) : \|\boldsymbol{y}\| \leq C, \|\boldsymbol{\zeta}\| \leq R\}$ with the balls $\cup_{\boldsymbol{y}} B(\boldsymbol{y}, 1/K_{\boldsymbol{y},\boldsymbol{\zeta}})$. By compactness, there is a finite subcover $(\boldsymbol{y}_1, \boldsymbol{\zeta}_1), \ldots, (\boldsymbol{y}_r, \boldsymbol{\zeta}_r)$, so we can take $K_R = \max_{i \in [r]} K_{\boldsymbol{y}_i, \boldsymbol{\zeta}_i} < \infty$ and the lemma holds since $\|\boldsymbol{y}\| \leq C$ almost surely by Assumption 2.1. $\qquad\square$

**Lemma D.6.** *The function $h(\boldsymbol{x}; \cdot)$ is locally bounded, Lipschitz and smooth in its second argument: for any $R > 0$ there exists $K_R$ such that for any $\boldsymbol{\psi}, \boldsymbol{\psi}' \in B(0, R)$,*

$$\|h(\boldsymbol{x}; \boldsymbol{\psi})\| \leq K_R$$
$$\|h(\boldsymbol{x}; \boldsymbol{\psi}) - h(\boldsymbol{x}; \boldsymbol{\psi}')\| \leq K_R \|\boldsymbol{\psi} - \boldsymbol{\psi}'\|$$
$$\|Dh(\boldsymbol{x}; \boldsymbol{\psi}) - Dh(\boldsymbol{x}; \boldsymbol{\psi}')\| \leq K_R \|\boldsymbol{\psi} - \boldsymbol{\psi}'\|,$$

*almost surely over $\boldsymbol{x}$. Here $Dh(\boldsymbol{x}, \cdot) \in \mathbb{R}^{d_{out}} \times R^p$ is the derivative in the second argument.*

*Proof.* Analogous to proof of Lemma D.5, using continuous twice-differentiability of $h$ and boundedness of $\|\boldsymbol{x}\|$. $\qquad\square$

**Lemma D.7** (Local Lipschitzness of loss and loss derivative). *When $\boldsymbol{\theta} = (\boldsymbol{u}, \boldsymbol{v}) \in \mathbb{R}^{2p}$ and $f_{\mathsf{NN}}(\boldsymbol{x}; \boldsymbol{\theta}) = h(\boldsymbol{x}; \boldsymbol{u} \odot \boldsymbol{u})$ the following holds for $\boldsymbol{g}(\boldsymbol{\theta})$ defined in (4). For any $R > 0$, there exists $K_R < \infty$ such that for any $\boldsymbol{\theta}, \boldsymbol{\theta}' \in B(0, K_R)$,*

$$\|\boldsymbol{g}(\boldsymbol{\theta}) - \boldsymbol{g}(\boldsymbol{\theta}')\| \leq K_R \|\boldsymbol{\theta} - \boldsymbol{\theta}'\|$$
$$\|\nabla_{\boldsymbol{\theta}} \mathcal{L}(\boldsymbol{\theta}) - \nabla_R \mathcal{L}(\boldsymbol{\theta}')\| \leq K_{\boldsymbol{\theta}} \|\boldsymbol{\theta} - \boldsymbol{\theta}'\|$$
$$|\mathcal{L}(\boldsymbol{\theta}) - \mathcal{L}(\boldsymbol{\theta}')| \leq K_R \|\boldsymbol{\theta} - \boldsymbol{\theta}'\|.$$

*Proof.* Let $\boldsymbol{\theta} = (\boldsymbol{u}, \boldsymbol{v}), \boldsymbol{\theta}' = (\boldsymbol{u}', \boldsymbol{v}')$. This follows immediately from the local Lipschitzness and smoothness of $h$ and $\ell$ in Lemmas D.5 and D.6, as well as

$$\|\boldsymbol{g}(\boldsymbol{\theta}) - \boldsymbol{g}(\boldsymbol{\theta}')\| = \|\mathbb{E}_{\boldsymbol{x},\boldsymbol{y}}[Dh(\boldsymbol{x}; \boldsymbol{u} \odot \boldsymbol{v})^\top D\ell(\boldsymbol{y}, h(\boldsymbol{x}; \boldsymbol{u} \odot \boldsymbol{v}))^\top - Dh(\boldsymbol{x}; \boldsymbol{u}' \odot \boldsymbol{v}')^\top D\ell(\boldsymbol{y}, h(\boldsymbol{x}; \boldsymbol{u}' \odot \boldsymbol{v}'))^\top]\|.$$
$$\square$$

