# Transformers learn through gradual rank increase

## Abstract

We identify incremental learning dynamics in transformers, where the difference between trained and initial weights progressively increases in rank. We rigorously prove this occurs under the simplifying assumptions of diagonal weight matrices and small initialization. Our experiments support the theory and also show that phenomenon can occur in practice without the simplifying assumptions.

## 1 Introduction

The transformer architecture achieves state of the art performance in various domains, yet we still lack a solid theoretical understanding of its training dynamics (Vaswani et al., 2017; Devlin et al., 2019; Liu et al., 2019; Dosovitskiy et al., 2020). Nevertheless, the theoretical toolbox has matured over the last years and there are promising new approaches. One important line of work examines the role that initialization scale plays on the trajectory taken by gradient descent (Jacot et al., 2018; Chizat et al., 2018; Geiger et al., 2019; Moroshko et al., 2020; Jacot et al., 2021; Stöger & Soltanolkotabi, 2021; Kim & Chung, 2022). When the weights are initialized small, it has been shown for simple networks that an *incremental learning* behaviour occurs, where functions of increasing complexity are learned in stages. This regime is known to be richer than the large-initialization regime[1], but the incremental learning dynamics are difficult to analyze, and are so far understood only for extremely simple architectures. Can we apply this analysis to transformers? Namely:

*Are there incremental learning dynamics when training a transformer architecture?*

An obstacle is that past work on incremental learning has mainly studied linear networks (Berthier, 2022; Arora et al., 2019; Milanesi et al., 2021; Li et al., 2020; Woodworth et al., 2019; Jacot et al., 2021; Gissin et al., 2019), with one paper studying nonlinear 2-layer fully-connected networks (Boursier et al., 2022). In contrast, transformers have nonlinear attention heads that do not fall under previous analyses: given $\boldsymbol{X} \in \mathbb{R}^{n \times d}$, an attention head computes

$$\text{attention}(\boldsymbol{X}; \boldsymbol{W}_K, \boldsymbol{W}_Q, \boldsymbol{W}_V, \boldsymbol{W}_O) = \text{smax}(\boldsymbol{X}\boldsymbol{W}_K\boldsymbol{W}_Q^\top\boldsymbol{X}^\top)\boldsymbol{X}\boldsymbol{W}_V\boldsymbol{W}_O^\top \tag{1}$$

where $\boldsymbol{W}_K, \boldsymbol{W}_Q, \boldsymbol{W}_V, \boldsymbol{W}_O \in \mathbb{R}^{d \times d'}$ are trainable matrices, and the softmax is applied row-wise. A transformer is even more complex, since it is formed by stacking alternating layers of attention heads and feedforward networks, along with residual connections.

**Main finding**   Our main finding is that transformers exhibit incremental learning dynamics, where *the difference between the trained and initial weights incrementally increases in rank*. Our results have a theoretical component and an experimental component.

---

[1] In the large-initialization regime, deep learning behaves as a kernel method Jacot et al. (2018); Chizat et al. (2018). Various separations with kernels are known for smaller initialization: e.g., Ghorbani et al. (2019); Abbe et al. (2022); Malach et al. (2021).

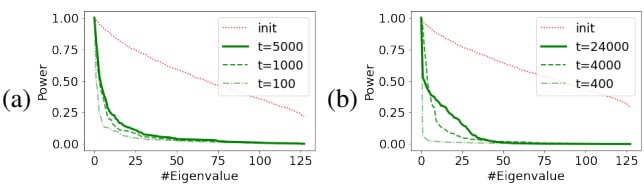

Figure 1: For an attention head in ViT trained on (a) CIFAR-10, and (b) ImageNet, we plot the normalized spectra of $\boldsymbol{W}_K \boldsymbol{W}_Q^\top$ at initialization (in red), and of the learned perturbations to $\boldsymbol{W}_K \boldsymbol{W}_Q^\top$ at different epochs (in green).

**Theoretical contributions** For our theory, we study a simplification of the transformer architecture, where the attention head weights are diagonal matrices: i.e., in each attention head we have $\boldsymbol{W}_K = \operatorname{diag}(\boldsymbol{w}_K)$, where $\boldsymbol{w}_K \in \mathbb{R}^d$ are trainable weights, and similarly for $\boldsymbol{W}_Q, \boldsymbol{W}_V$ and $\boldsymbol{W}_O$. We rigorously establish the training dynamics of this architecture under gradient flow when the initialization is small. We prove that dynamics occur in discrete stages: (1) during most of each stage, the loss plateaus because the weights remain close to a saddle point, and (2) at the end, the saddle point is quickly escaped and the rank of the weights increases by at most one.

This theoretical result on transformers follows from a general theorem characterizing the learning dynamics of networks $f_{\mathsf{NN}}$ that depend on the product of parameters $\boldsymbol{u}, \boldsymbol{v} \in \mathbb{R}^p$ as

$$f_{\mathsf{NN}}(\boldsymbol{x}; \boldsymbol{u}, \boldsymbol{v}) = h(\boldsymbol{x}; \boldsymbol{u} \odot \boldsymbol{v}), \tag{2}$$

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

Here $D\ell(\boldsymbol{y}, \cdot) \in \mathbb{R}^{1 \times d_{out}}$ is the derivative of $\ell$ in the second argument and $Dh(\boldsymbol{x}, \cdot) \in \mathbb{R}^{d_{out} \times p}$ is the derivative of $h$ in the second argument. We show that if initialization scale of $\boldsymbol{\theta} = (\boldsymbol{u}, \boldsymbol{v})$ is small, then learning proceeds in incremental stages, as given in Algorithm 1, where in each stage the effective sparsity of $\boldsymbol{u}$ and $\boldsymbol{v}$ increases by at most one.

### 4.1 Intuition for incremental learning dynamics

We develop an informal intuition for the result. First, we observe a conservation law that simplifies the dynamics. It can be viewed as the balancedness property for networks with linear activations Arora et al. (2018); Du et al. (2018), specialized to the case of diagonal layers.

**Lemma 4.1** (Conservation law). *For any $i \in [p]$ and any time $t$, we have*
$$u_i^2(t) - v_i^2(t) = u_i^2(0) - v_i^2(0). \tag{5}$$

*Proof.* This follows from $\frac{d}{dt}(u_i^2 - v_i^2) = u_i v_i g_i(\boldsymbol{\theta}) - u_i v_i g_i(\boldsymbol{\theta}) = 0$. $\qquad \square$

This reduces the degrees of freedom and means that we need only keep track of $p$ parameters in total. Specifically, if we define $w_i(t) := u_i(t) + v_i(t)$, then the vector $\boldsymbol{w} = \boldsymbol{u} + \boldsymbol{v}$ evolves by
$$\frac{d\boldsymbol{w}}{dt} = \boldsymbol{w} \odot \boldsymbol{g}(\boldsymbol{\theta}). \tag{6}$$
Using the conservation law (5), one can compute $\boldsymbol{\theta}(t)$ from $\boldsymbol{w}(t)$, so it remains to analyze the dynamics of $\boldsymbol{w}(t)$.

### 4.1.1 Stage 1 of dynamics

**Stage 1A of dynamics: loss plateau for time** $\Theta(\log(1/\alpha))$    At very early times $t$, we have $\boldsymbol{\theta}(t) \approx \mathbf{0}$ because the weights are initialized to be very small. Thus, we can approximate $\boldsymbol{g}(\boldsymbol{\theta}(t)) \approx \boldsymbol{g}(\mathbf{0})$ and so we can solve for the evolution of $\boldsymbol{w}$:

$$\boldsymbol{w}(t) \approx \boldsymbol{w}(0) \odot e^{\boldsymbol{g}(\mathbf{0})t}.$$

This approximation is valid until one of the entries of $\boldsymbol{\theta}(t)$ reaches constant size, which one can show happens around time $t \approx T_1 \cdot \log(1/\alpha)$ for

$$T_1 = \min_{i \in [p]} 1/|g_i(\mathbf{0})|\,.$$

Until this time, the weights $\boldsymbol{\theta}(t)$ are small, the network remains close to its initialization, and so we observe a loss plateau.

**Stage 1B of dynamics: nonlinear dynamics for time** $O(1)$    Subsequently, we observe a rapid decrease of the loss and nonlinear dynamics during a $O(1)$-order time-scale. Indeed, suppose that the dynamics are "non-degenerate" in the sense that there is a unique coordinate $i_0$ such that $1/|g_{i_0}(\mathbf{0})| = T_1$. Under this assumption, in stage 1A, the weights only grow significantly at coordinate $i_0$. So one can show that for any small $\epsilon > 0$, there is a time $\underline{t}_1(\epsilon) \approx T_1 \cdot \log(1/\alpha)$ such that $u_{i_0}(\underline{t}_1) \approx \epsilon$, $v_{i_0}(\underline{t}_1) \approx s\epsilon$ for some sign $s \in \{+1, -1\}$, and $|u_i(\underline{t}_1)|, |v_i(\underline{t}_1)| = o_\alpha(1)$ for all $i \neq i_0$.[3]

Because all coordinates except for $i_0$ are negligibly small after stage 1A, we may perform the following approximation of the dynamics. Zero out the weights at coordinates except for $i_0$, and consider the training dynamics starting at $\tilde{\boldsymbol{\theta}} = (\epsilon \boldsymbol{e}_{i_0}, s\epsilon \boldsymbol{e}_{i_0})$. After some constant time, independent of $\alpha$, these dynamics should approach a stationary point. Furthermore, all coordinates of $\boldsymbol{u}$ and $\boldsymbol{v}$ will remain zero except for the $i_0$ coordinate, so the sparsity of the weights will be preserved. In other words, we should expect there to be a time $\bar{t}_1 = \underline{t}_1 + O(1) \approx T_1 \cdot \log(1/\alpha)$ such that

$$\boldsymbol{\theta}(\bar{t}_1) \approx (a\boldsymbol{e}_{i_0}, sa\boldsymbol{e}_{i_0}) := \boldsymbol{\theta}^1\,,$$

for some $a \in \mathbb{R}_{>0}$, such that $\boldsymbol{\theta}^1$ is a stationary point of the loss.[4] This is a good approximation because $\bar{t}_1 - \underline{t}_1 = O(1)$ is a constant time-scale, so the weights at coordinates except for $i_0$ remain negligible between times $\underline{t}_1$ and $\bar{t}_1$. Overall, we have argued that the network approximately reaches stationary point that is 1-sparse, where only the weights at coordinate $i_0$ are nonzero.

### 4.1.2 Later stages

We can extend the argument to any number of stages $k$, where in each stage the weights remain close to constant for time $\Theta(\log(1/\alpha))$ and then rapidly change during time $O(1)$, with the sparsity of the weights increasing by at most one. In order to analyze multiple stages, we must also keep track of the magnitude of the weights on the logarithmic scale because these evolve nonnegligibly throughout training. Inductively on $k$, suppose that there is some $T_k \in \mathbb{R}, \boldsymbol{b}^k \in \mathbb{R}^p$ and $\boldsymbol{\theta}^k \in \mathbb{R}^{2p}$ and a time $\bar{t}_k \approx T_k \cdot \log(1/\alpha)$ such that

$$\log_\alpha(\boldsymbol{w}(\bar{t}_k)) \approx \boldsymbol{b}^k \text{ and } \boldsymbol{\theta}(\bar{t}_k) \approx \boldsymbol{\theta}^k,$$

where $\boldsymbol{\theta}^k$ is a stationary point of the loss. We argue for the inductive step that there is $T_{k+1} \in \mathbb{R}$ such that during times $t \in (\bar{t}_k, T_{k+1} \cdot \log(1/\alpha) - \Omega(1))$ the weights remain close to the stationary point from the previous phase, i.e., $\boldsymbol{\theta}(t) \approx \boldsymbol{\theta}^k$. And at a time $\bar{t}_{k+1} \approx T_{k+1} \cdot \log(1/\alpha)$ we have

$$\log_\alpha(\boldsymbol{w}(\bar{t}_{k+1})) \approx \boldsymbol{b}^{k+1} \text{ and } \boldsymbol{\theta}(\bar{t}_{k+1}) \approx \boldsymbol{\theta}^{k+1},$$

where $\boldsymbol{\theta}^{k+1}$ and $\boldsymbol{b}^{k+1}$ are defined below, and $\boldsymbol{\theta}^{k+1}$ is a stationary point of the loss whose support has grown by at most one compared to $\boldsymbol{\theta}^k$. The pseudocode for the evolution of $\boldsymbol{b}^k$ and $\boldsymbol{\theta}^k$ along the stages is given in Algorithm 1, and more details are provided below.

---

[3]Without loss of generality, we can ensure that at initialization $\boldsymbol{u}(0)$ and $\boldsymbol{u}(0) + \boldsymbol{v}(0)$ are nonnegative. This implies $\boldsymbol{u}(t)$ is nonnegative. The fact that $u_{i_0}$ and $v_{i_0}$ are roughly equal in magnitude but might differ in sign is due to the conservation law (5). See Appendix A.3 for details.

[4]The entries of $\boldsymbol{u}$ and $\boldsymbol{v}$ are close in magnitude (but may differ in sign) because of the conservation law (5).

**Stage** $(k+1)$**A, loss plateau for time** $\Theta(\log(1/\alpha))$  At the beginning of stage $k+1$, the weights are close to the stationary point $\boldsymbol{\theta}^k$, and so, similarly to stage 1A, linear dynamics are valid.

$$\boldsymbol{w}(t) \approx \boldsymbol{w}(\bar{t}_k) \odot e^{\boldsymbol{g}(\boldsymbol{\theta}^k)(t-\bar{t}_k)} \,. \tag{7}$$

Using the conservation law (5), we derive a "time until active" for each coordinate $i \in [p]$, which corresponds to the time for the weight at that coordinate to grow from negligible to nonnegligible magnitude:

$$\Delta_k(i) = \begin{cases} (b_i^k - 1 + \mathrm{sgn}(g_i(\boldsymbol{\theta}^k)))/g_i(\boldsymbol{\theta}^k), & \text{if } g_i(\boldsymbol{\theta}^k) \neq 0 \\ \infty, & \text{if } g_i(\boldsymbol{\theta}^k) = 0 \end{cases} \tag{8}$$

The approximation (7) therefore breaks down at a time $t \approx T_{k+1} \cdot \log(1/\alpha)$, where

$$T_{k+1} = T_k + \Delta_k(i_k), \quad i_k = \arg\min_{i\in[p]} \Delta_k(i) \,, \tag{9}$$

which corresponds to the first time at the weights at a coordinate grow from negligible to nonnegligible magnitude. And at times $t \approx T_{k+1} \cdot \log(1/\alpha)$, on the logarithmic scale $\boldsymbol{w}$ is given by

$$\log_\alpha(\boldsymbol{w}(t)) \approx \boldsymbol{b}^{k+1} := \boldsymbol{b}^k - \boldsymbol{g}(\boldsymbol{\theta}^k)\Delta_k(i_k) \,, \tag{10}$$

**Stage** $(k+1)$**B of dynamics: nonlinear dynamics for time** $O(1)$  Subsequently, the weights evolve nonlinearly during $O(1)$ time. To see this, if we make the non-degeneracy assumption that there is a unique coordinate $i_k$ such that $\Delta_k(i_k) = \min_i \Delta_k(i)$, then this means that in stage $(k+1)$A, the only coordinate where weights grow from negligible to nonnegligible magnitude is $i_k$. Roughly speaking, for any $\epsilon > 0$, there is a time $\underline{t}_{k+1}(\epsilon) \approx T_{k+1} \cdot \log(1/\alpha)$ such that

$$\boldsymbol{\theta}(\underline{t}_{k+1}) \approx \boldsymbol{\theta}^k + (\epsilon \boldsymbol{e}_{i_k}, \mathrm{sgn}(g_i(\boldsymbol{\theta}^k))\epsilon \boldsymbol{e}_{i_k}) \,,$$

where the sign of the weights in coordinate $i_k$ comes from the conservation law (5). At this time, the weights are approximately the stationary point from stage $k$, plus a small perturbation. Consider the dynamics of $\boldsymbol{\psi}^k(t,\epsilon) \in \mathbb{R}^{2p}$ initialized at $\boldsymbol{\psi}^k(0,\epsilon) = \boldsymbol{\theta}^k + (\epsilon \boldsymbol{e}_{i_k}, \mathrm{sgn}(g_i(\boldsymbol{\theta}^k))\epsilon \boldsymbol{e}_{i_k})$ and evolving according to the gradient flow $\frac{d\boldsymbol{\psi}^k(t,\epsilon)}{dt} = -\nabla_{\boldsymbol{\theta}}\mathcal{L}(\boldsymbol{\psi}^k)$. These dynamics may be highly nonlinear, so to control them let us assume that as we take $\epsilon$ to be small, they converge to a limiting point $\boldsymbol{\theta}^{k+1}$

$$\lim_{\epsilon\to 0}\lim_{t\to\infty} \boldsymbol{\psi}^k(t,\epsilon) = \boldsymbol{\theta}^{k+1} \,. \tag{11}$$

Then we expect that at a time $\bar{t}_{k+1} = \underline{t}_{k+1} + O(1) \approx T_{k+1} \cdot \log(1/\alpha)$, we have $\boldsymbol{\theta}(\bar{t}_{k+1}) \approx \boldsymbol{\theta}^{k+1}$. This concludes the inductive step.

## 4.2 Formal statement of incremental dynamics

We formally state our result. For ease of notation, we write $\boldsymbol{\theta}^k = (\boldsymbol{u}^k, \boldsymbol{v}^k)$ and $\boldsymbol{v}^k = \boldsymbol{s}^k \odot \boldsymbol{u}^k$ for some sign-flip vector $\boldsymbol{s}^k \in \{+1,-1\}^k$. This form of $\boldsymbol{\theta}^k$ can be guaranteed by the conservation law (5) of the dynamics; see Appendix A. We also denote $\mathrm{supp}(\boldsymbol{\theta}^k) := \mathrm{supp}(\boldsymbol{u}^k) = \mathrm{supp}(\boldsymbol{v}^k) \subseteq [p]$.

We state our assumptions formally. First, we require that the dynamics be non-degenerate, in the sense that two coordinates do not become active at the same time. We also place a technical condition to handle the corner case when a coordinate leaves the support of active coordinates.

---

**Algorithm 1** Incremental learning in networks with diagonal weights

---
1: $\boldsymbol{b}^0, \boldsymbol{\theta}^0 \leftarrow \boldsymbol{0} \in \mathbb{R}^p$, $T_0 \leftarrow 0$
2: **for** stage number $k = 0, 1, 2, \ldots$ **do**
3:     # (A) Pick new coordinate $i_k \in [p]$ to activate.
4:     For each $i$, define time $\Delta_k(i)$ until active using (8).
5:     Pick winning coordinate $i_k$ using (9)
6:     Calculate time $T_{k+1}$ using (9) and **break if** $\infty$
7:     Update logarithmic weight approximation $\boldsymbol{b}^{k+1}$ using (10)
8:     # (B) Train activated coordinates to stationarity.
9:     $\boldsymbol{\theta}^{k+1} \leftarrow$ limiting dynamics point from (11)
10: **end for**

---

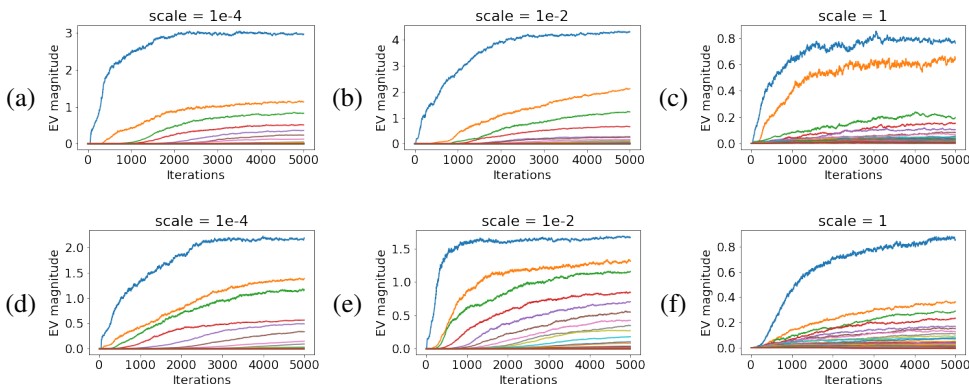

Figure 2: Training a vision transformer on CIFAR-10 using Adam, while varying the initialization scale (unit scale indicates default initialization). Plotted are the evolution of the eigenvalues of $\Delta \boldsymbol{W}_K \boldsymbol{W}_Q^\top$ (a) - (c) and $\Delta \boldsymbol{W}_V \boldsymbol{W}_O^\top$ (d) - (f) in a random self-attention head in the second layer throughout training. Incremental learning dynamics and a low-rank bias are evident for all scales, albeit more pronounced at smaller initialization scales.

**Assumption 4.2** (Nondegeneracy of dynamics in part (A))**.** The initialization satisfies $u_i(0) \neq v_i(0)$ for all $i$. For stage $k$, either $T_k = \infty$ or there is a unique minimizer $i_k$ to $\min_i \Delta_k(i_k)$ in (9). Finally, for all $i \in \operatorname{supp}(\boldsymbol{\theta}^{k-1}) \setminus \operatorname{supp}(\boldsymbol{\theta}^k)$ we have $g_i(\boldsymbol{\theta}^k) \neq 0$.

Next, we require that very small perturbations of the coordinates outside of $\operatorname{supp}(\boldsymbol{\theta}^k)$ do not change the dynamics. For this, it suffices that $\boldsymbol{\theta}^k$ be a strict local minimum.

**Assumption 4.3** (Stationary points are strict local minima)**.** For stage $k$, there exist $\delta_k > 0$ and $c_k > 0$ such that for $\tilde{\boldsymbol{u}} \in B(\boldsymbol{u}^k, \delta)$ supported on $\operatorname{supp}(\boldsymbol{u}^k)$, we have

$$\mathcal{L}(\tilde{\boldsymbol{u}}, \boldsymbol{s}^k \odot \tilde{\boldsymbol{u}}) \geq c_k \|\boldsymbol{u}^k - \tilde{\boldsymbol{u}}\|^2$$

Finally, we require a robust version of the assumption (11), asking for convergence to a neighborhood of $\boldsymbol{\theta}^{k+1}$ even when the initialization is slightly noisy.

**Assumption 4.4** (Noise-robustness of dynamics in part (B))**.** For any stage $k$ with $T_{k+1} < \infty$ and any $\epsilon > 0$, there are $\delta > 0$ and $\tau : \mathbb{R}_{>0} \to \mathbb{R}$ such that the following holds. For any $\tilde{\boldsymbol{u}} \in B(\boldsymbol{u}^k, \delta) \cap \mathbb{R}^p_{\geq 0}$ supported on $\operatorname{supp}(\tilde{\boldsymbol{u}}) \subseteq \operatorname{supp}(\boldsymbol{u}^k) \cup \{i_k\}$, there exists a unique solution $\boldsymbol{\psi} : [0, \infty) \to \mathbb{R}^p$ of the gradient flow $\frac{d\boldsymbol{\psi}}{dt} = -\nabla_{\boldsymbol{\theta}} \mathcal{L}(\boldsymbol{\psi})$ initialized at $\boldsymbol{\psi}(0) = (\tilde{\boldsymbol{u}}, \boldsymbol{s}^{k+1} \odot \tilde{\boldsymbol{u}})$, and at times $t \geq \tau(\tilde{\psi}_{i_k})$,

$$\|\boldsymbol{\psi}(t) - \boldsymbol{\theta}^{k+1}\| < \epsilon \,.$$

These assumptions are validated experimentally in Appendix C. Using them, we prove that incremental learning Algorithm 1 tracks the gradient flow dynamics if the initialization scale is small.

**Theorem 4.5** (Incremental dynamics with untied weights)**.** *For any stage $k$ and time $t \in (T_k, T_{k+1})$ the following holds under Assumptions 4.2 4.3 and 4.4. There is $\alpha_0(t) > 0$ such that for all $\alpha < \alpha_0$, there exists a unique solution $\boldsymbol{\theta} : [0, t \log(1/\alpha)] \to \mathbb{R}^p$ to the gradient flow (3) and*

$$\lim_{\alpha \to 0} \boldsymbol{\theta}(t \cdot \log(1/\alpha)) \to \boldsymbol{\theta}^k \,,$$

*and at each stage the sparsity increases by at most one:* $\operatorname{supp}(\boldsymbol{\theta}^{k+1}) \setminus \operatorname{supp}(\boldsymbol{\theta}^k) \subseteq \{i_k\}$.

**Example 4.6** (Application: Incremental learning in diagonal transformer)**.** *In Example 3.2, we showed that a diagonal transformer falls under Theorem 4.5. As a corollary, the gradient flow on a transformer with small initialization will learn in stages, where in each stage there will be at most one head $i \in [H]$ on one layer $\ell \in [L]$ such that either the rank of $\boldsymbol{W}_K^{\ell,i}(\boldsymbol{W}_Q^{\ell,i})^\top = \operatorname{diag}(\boldsymbol{w}_K^{\ell,i})\operatorname{diag}(\boldsymbol{w}_Q^{\ell,i})$ or the rank of $\boldsymbol{W}_V^{\ell,i}(\boldsymbol{W}_O^{\ell,i})^\top = \operatorname{diag}(\boldsymbol{w}_V^{\ell,i})\operatorname{diag}(\boldsymbol{w}_O^{\ell,i})$ increases by at most one.*

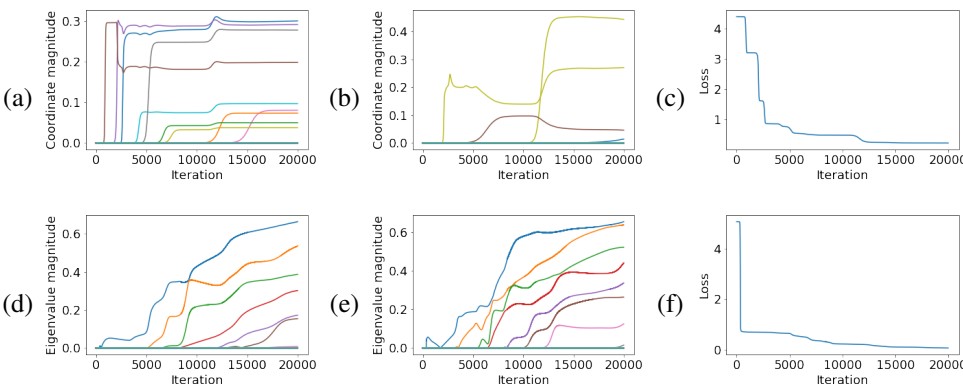

Figure 3: A network containing a single self-attention layer with diagonal (a) - (c) and full (d) - (f) weight matrices, trained with gradient descent in the incremental learning regime. (a) The diagonal entries of $\boldsymbol{W}_V \boldsymbol{W}_O^\top$ and (d) the singular values of $\boldsymbol{W}_V \boldsymbol{W}_O^\top$ are learned incrementally. (b) The diagonal entries of $\boldsymbol{W}_K \boldsymbol{W}_Q^\top$ and (e) the singular values of $\boldsymbol{W}_K \boldsymbol{W}_Q^\top$ are learned incrementally. (c), (f) The loss curves show stagewise plateaus and sharp decreases.

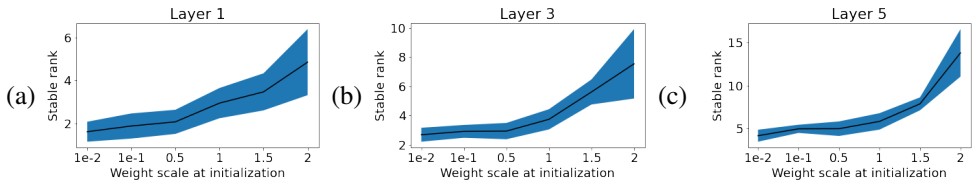

Figure 4: Stable rank of $\Delta \boldsymbol{W}_K \boldsymbol{W}_Q^\top$ per initialization scale (Unit scale refers to the default initialization) in different self-attention heads post-training, at layers 1, 3, 5. At each layer, the stable rank mean and standard deviation are computed across 8 heads per layer, for each initialization scale. All models were trained on CIFAR-10 using the Adam optimizer. Smaller initialization scales lead to lower-rank attention heads. Analogous plots for $\Delta \boldsymbol{W}_V \boldsymbol{W}_O^\top$ are in the appendix.

## 5 Experimental results

We experimentally support our theoretical findings in a series of experiments: first on a toy model given by Equation (1), followed by experiments on a vision transformer on the CIFAR datasets. We defer additional experimental details and results to the appendix.

**Toy models** We consider a toy model comprised of one self-attention layer with a single head as in (1), with either diagonal or full weight matrices. We initialize $\boldsymbol{W}_K, \boldsymbol{W}_Q, \boldsymbol{W}_V, \boldsymbol{W}_O$ using Gaussian initialization with a small standard deviation, and train the model using GD on a regression task with 50-dimensional random Gaussian token inputs and targets from a teacher model. During training, we track the diagonal entries of $\boldsymbol{W}_K \boldsymbol{W}_Q^\top$ and $\boldsymbol{W}_V \boldsymbol{W}_O^\top$ in the diagonal case, and the singular values of $\boldsymbol{W}_K \boldsymbol{W}_Q^\top$ and $\boldsymbol{W}_V \boldsymbol{W}_O^\top$ in the full weights case. Our results are summarized in Figure 3. For the diagonal model, as predicted, diagonal components are learned incrementally, resulting in progressive increase in the rank; in Appendix C we run additional experiments to verify that the assumptions of Theorem 4.5 indeed hold. For the full-weights model, we also observe incremental learning with progressively-increasing rank, even though this setting falls beyond our theory.

**Vision transformers** We next run experiments that go well beyond our toy model to test the extent to which incremental learning with a low-rank bias exists in popular models used in practice. We conduct experiments with vision transformers (ViT) Dosovitskiy et al. (2020) trained on the CIFAR-10/100 and ImageNet datasets. We use a ViT of depth 6, with 8 self-attention heads per layer (with layer normalization). We use an embedding and MLP dimension of $d_{\text{emb}} = 512$, and a head dimension of $d_h = 128$ (i.e $\boldsymbol{W}_K, \boldsymbol{W}_Q, \boldsymbol{W}_V, \boldsymbol{W}_O \in \mathbb{R}^{d_{\text{emb}} \times d_h}$). We train the transformer using Adam

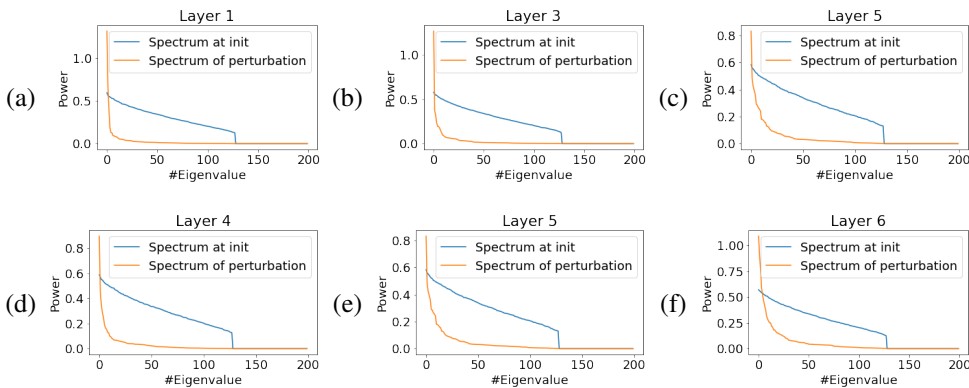

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

652 $\qquad\square$

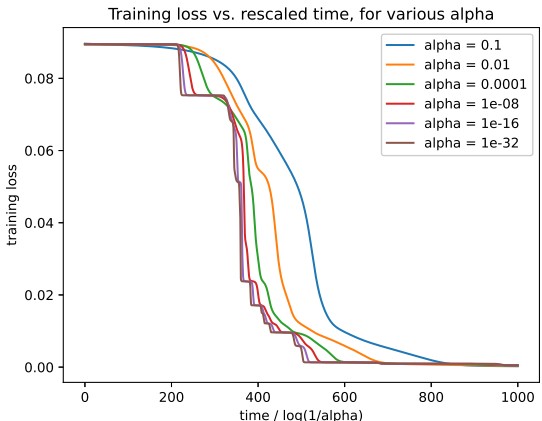

Figure 6: Evolution of loss versus rescaled time initializing at various scalings $\alpha$ in the toy task of learning an attention head with diagonal weights. The loss curves converge as $\alpha \to 0$ to a curve with loss plateaus and sharp decreases, as predicted by the theory.

## C    Experimental validation of the assumptions in Theorem 4.5

In Figures 6, 7, and 8, we plot the evolution of the losses, of the entries of $\boldsymbol{W}_K \boldsymbol{W}_Q^\top = \mathrm{diag}(\boldsymbol{w}_K)\mathrm{diag}(\boldsymbol{w}_Q)$, and of the entries of $\boldsymbol{W}_V \boldsymbol{W}_O^\top = \mathrm{diag}(\boldsymbol{w}_V)\mathrm{diag}(\boldsymbol{w}_O)$ in the toy task of training an attention head (1) with diagonal weights. The model is trained with SGD on the mean-squared error loss on 1000 random samples $(\boldsymbol{X}, \boldsymbol{y})$. Each random sample has $\boldsymbol{X} \in \mathbb{R}^{10 \times 50}$, which a sequence of 10 tokens, each of dimension 50, which is distributed as isotropic Gaussians. The label $\boldsymbol{y}$ is given by a randomly-generated teacher model that is also an attention head (1) with diagonal weights. In Figures 6, 7, and 8, for $\alpha \in \{0.1, 0.01, 0.0001, 10^{-8}, 10^{-16}, 10^{-32}\}$ we plot the evolution of the loss and of the weights when initialized at $\boldsymbol{\theta}(0) = \alpha \boldsymbol{\theta}_0$, for some random Gaussian $\boldsymbol{\theta}_0$. Qualitatively, as $\alpha \to 0$ we observe that the loss curve and the trajectories of the weights appear to converge to a limiting stagewise dynamics, where there are plateaus followed by movement on short time-scales, as predicted by the theory.

**Validation of Assumption 4.2 (non-degeneracy of dynamics)**    As $\alpha \to 0$, notice that the stages appear to separate and happen at distinct times. Furthermore, at no stage do any of the nonnegligible coordinates leave the support of $\boldsymbol{\theta}$, so