# OpenReview forum: "Transformers learn through gradual rank increase"
_NeurIPS.cc/2023/Conference — NeurIPS 2023 poster_

### Official Review · Reviewer_yLKb · 2023-07-06

**Soundness:** 4 excellent
**Presentation:** 3 good
**Contribution:** 3 good
**Rating:** 6
**Confidence:** 3

**Summary:**

This paper presents theoretical justifications and empirical evidence that transformers demonstrate incremental learning dynamics in the low-initialization regime. The authors consider a very restricted diagonal attention model along with a range of restrictive assumptions to theoretically characterize a number of features of learning dynamics in single-layer transformers (a single attention layer). The assumptions, which include nondegeneracy of dynamics (4.2), existence of stationary points that are strict local minima (4.3), and robustness of gradient flow to perturbations (4.4). These assumptions, in the diagonal model, allow the authors to prove that learning evolves through discrete, incremental stages of gradual rank increase. These assumptions and theoretical predictions are experimentally verified using a toy learning scenario matching the assumptions, and results are given on a full multi-layer, multi-head vision transformer which demonstrate similar learning dynamics despite being significantly far from the restrictive assumptions of the theory.


**Strengths:**

1. **Convincing Theory**. The theoretical framework and proof of incremental learning dynamics is fairly easy to follow and, although based on a very restricted model and stringent assumptions, I think provides a number of useful tools of understanding and studying the learning dynamics of more complex models.
2. **Fairly Convincing Empirical Confirmation**. The empirical validation of incremental dynamics in the single attention layer model (Figure 3) are interesting -- in both the diagonal and full model the empirically observed behavior follows the predicted incremental "activation" of directions. However I find it odd that the evolution of singular values is so markedly different between the value-output and key-query matrices (Figures 3a-d and 3b-e). The value-output eigenvalues are significantly more sparse and their activations more "instantaneous". This difference is also roughly observable in the ViT results in Figure 2 (although reversed!).


**Weaknesses:**

1. **Organization**. As will many theoretical papers, some of the most interesting observations are relegated to the Supplementary Material -- which I appreciate is inevitable. Reorganizing the preliminaries and theoretical development of the main paper I admit is delicate, but I think that the stable rank analysis in Figure 4 and illustration of low-rank bias in Figure 5 add little to the empirical results section of the main paper. Instead, I think this space could have been used better providing experimental evidence of the assumptions 4.2, 4.3, and 4.4 (is there an assumption 4.1 that I am missing?). Some of the plots in Figures 9 and 10 of the Supplementary Material, or even the *very* clear illustration of stepped behavior and dependence on $\alpha$ from Figure 6 would be excellent illustrations to include in the main paper. I find the plots in rescaled training time much easier to interpret.
2. **ViT Training Regime**. All of the empirical results on ViT were generated using Adam, which I fear might introduce its own dynamics into the learning process due to its modulation of gradients using the diagonal empirical Fisher matrix. It would be interesting to see results generated using vanilla SGD to verify that the observed learning dynamics are due to the low-initialization regime.


**Questions:**

1. Why is the rank at initialization bounded (at 128?) in the plots in Figure 5?
2. Is the top/bottom organization of plots indicated in the captions of Figures 2 and 3 correct? If so, can you explain the difference between behavior between the key-query and value-output eigenvalues?
3. Are the same learning dynamics observed in ViT when training with SGD as opposed to Adam?


**Limitations:**

The authors provide a discussion of the limitations of the results presented in the paper, acknowledging that the restrictive assumptions and requirements of the theory are quite far from actual practice. They also connect their to recent work on low-rank model adaptation which could potentially exploit the theoretical and empirically observed incremental learning dynamics of transformers.

---

> ### Author Rebuttal · Authors · 2023-08-09
>
> Thank you for your positive review and for your helpful suggestions on presentation and paper organization. We are happy that you found our theory and experiments convincing. We answer your questions below.
>
> * Q1: “Why is the rank at initialization bounded (at 128?) in the plots in Figure 5?“
>     * The rank is bounded because the head dimension is 128 and embedding dimension is 512, meaning that the matrices $W_Q, W_K, W_V, W_O$ are in $\\mathbb{R}^{512 \\times 128}$.
> * Q2: “Is the top/bottom organization of plots indicated in the captions of Figures 2 and 3 correct? If so, can you explain the difference between behavior between the key-query and value-output eigenvalues?”
>     * Yes, the top/bottom organization is correct, but it is not consistent between Figure 2 and Figure 3, which may have caused confusion. We will make it consistent in the revision. Both plots demonstrate that incremental learning dynamics occur for both key-query and value-output matrices.
>     * You make an interesting point that there seem to be some qualitative differences in the evolution of keys-queries and values-outputs matrices. See also Figures 12, 14 where the stable-rank of $\\Delta W_VW_O^T$ is higher than that of the $\\Delta W_Q W_K^T$ for CIFAR-10 and CIFAR-100. However, this trend does not hold for ImageNet (Figure 16). Our current theory unfortunately does not give any clue for why $\\Delta W_VW_O^T$ might have higher stable-rank than $\\Delta W_QW_K^T$, or vice-versa.
> * Q3: “Are the same learning dynamics observed in ViT when training with SGD as opposed to Adam?“
>     * Yes. Thanks for the suggestion. We have added experiments on SGD confirming this. See attached document.
> * Thank you also for your excellent suggestions on organization. We agree that moving Figure 6 and some panels from Figures 9 and 10 to the main text will help readability of Section 4, both for illustrating the theorem statement and illustrating and justifying the assumptions. We will implement this in the revision.

---

> > ### Comment · Reviewer_yLKb · 2023-08-16
> >
> > Many thanks for your responses to my main questions, curiosities and concerns. The responses address all of my concerns and my opinion of this work after reading the rebuttal and the other reviews remains positive. If I interpret the results from Figures 2 and 3 in the Rebuttal PDF, the results using SGD not only exhibit the same rank-increasing trend, they are also quite a bit more stable compared to using Adam. I encourage the authors to include these observations, as well as the results on training GPT-2 provided in response to Reviewer ZvU1 in any final version of this work.

---

### Official Review · Reviewer_1Eo1 · 2023-07-08

**Soundness:** 4 excellent
**Presentation:** 3 good
**Contribution:** 3 good
**Rating:** 6
**Confidence:** 3

**Summary:**

This paper conducts solid analysis and experiments to demonstrate the theory and proofs provided in the paper offer valuable insights into the incremental learning dynamics in transformers and how they can be better understood.

**Strengths:**

1. The theory and proofs provided in the paper offer valuable insights into the incremental learning dynamics in transformers and how they can be better understood.
2. The experiments conducted in the paper also provide evidence to support the proposed approach.

**Weaknesses:**

Besides LoRA, are there any other methods and techniques related to the proof and conclusions of this paper? A further discussion may help.

**Questions:**

Please refers to the point listed in the Weakness part.

**Limitations:**

The proofs and conclusions of this study are dependent on the assumptions of the diagonal weights and small initialization. Such limitations have been discussed in an entire paragraph of Section 6.

---

> ### Author Rebuttal · Authors · 2023-08-10
>
> Thank you for your positive evaluation of our paper. We were happy to read that you thought our contribution provided valuable insights and that our experiments gave good evidence. We answer your question below:
>
> * Q1: Besides LoRA, are there any other methods and techniques related to the proof and conclusions of this paper? A further discussion may help.
>   * Training dynamics of transformers are a very active area of research, and so there are indeed other methods and techniques related to the ideas in this paper that have appeared recently.
>       * A very relevant paper called “InRank: Incremental Low-Rank Learning” [Zhao et al. ’23] appeared on arXiv in June, after the NeurIPS deadline. Similarly to us, the authors observe gradual rank increase dynamics during transformer training. This inspires [Zhao et al. ’23] to run LoRA to train a transformer starting at random initialization, and they obtain significant runtime/memory improvements over regular training, with little loss in performance. The theory in [Zhao et al. ’23] is incomparable to ours — and actually it is quite complementary. Their theory studies linear networks with orthogonal weights (which are equivalent to linear diagonal networks because the modes evolve separately) in standard initialization scale regimes. In contrast, in our theory we study nonlinear networks with diagonal weights in small initialization scale regimes.
>       * A recent ICML paper called “On the stepwise nature of self-supervised learning” analyzes the gradient flow dynamics of a simple model and an SSL objective. They reach a similar conclusion where progress is made in discrete steps, where the rank of the learned embeddings increase gradually. The analysis conducted in that paper differs from ours in that they use a linear model as a theoretical testbed, where we consider a nonlinear model.
>
>
> * Limitations. We agree that our theory has these limitations. But, for completeness, we recall that our experiments on ViTs and GPT2 are a contribution, and these do not depend on diagonal weights and small initialization.
> * References
>     * Zhao, Jiawei, et al. "InRank: Incremental Low-Rank Learning." arXiv preprint arXiv:2306.11250 (2023).
>     * Simon, James B., et al. "On the stepwise nature of self-supervised learning." arXiv preprint arXiv:2303.15438 (2023).

---

### Official Review · Reviewer_ZvU1 · 2023-07-09

**Soundness:** 2 fair
**Presentation:** 1 poor
**Contribution:** 2 fair
**Rating:** 4
**Confidence:** 3

**Summary:**

In this paper, the authors study the learning dynamics of transformers and argued that the difference between weights and their initial values increase in rank as the training progresses. Under small initialization, smoothness, non-degeneracy. convergence and robustness assumptions, the authors proof the incremental dynamics in diagonal-weighted transformers. Empirical results on toy models and VITs shows similar rank-progression dynamics on learned perturbations.

**Strengths:**

The authors make an interesting observation that the learned perturbations in transformers are of low rank and exhibit an rank-increasing dynamics. Theoretical analysis is given under a simplified setting and the empirical results seem to support the observation.
I do think this is a topic worth exploring as the findings could support the recently popular low ranking fine-tuning studies and potentially help identifying more efficient low-rank training methods through better understanding the learning dynamics.

**Weaknesses:**

* The assumptions are very strong such that it is not clear if the theoretical results has real implications in practical cases. Not only does the theory only hold for attention layer only, diagonal weighted transformers, it also need strong assumptions on convergence, robustness.

* The presentation of the paper could be improved. Section 4.1.1 and 4.1.2 are kind of confusing and does not help in understanding the reasoning behind the rank increasing dynamics. In addition, the author uses a lot of vague expressions like "very small", "non-negligible", "good approximation", when not providing quantitative reasoning like error/magnitude bounds.

* Dynamic (3) seems to hold only for gradient descent training, which is hardly used in practice. It's not clear if the analysis holds for optimizers like SGD/Adam.

**Questions:**

* Why is the non-toy experiments only conducted on VIT? Is there any reason to choose only the vision transformers but not regular NLP transformers?

* How small is a "small initialization" in practice?

* Dynamic (3) seems to hold only for gradient descent training, how does it apply to other optimizers like SGD/Adam?

* The learning dynamics theories in this paper is proved in continuous setting (learning rate $\rightarrow$ 0, learning steps $\rightarrow\infty$). The real discretized GD training is just a forward finite difference approximation of the dynamics (3). Under what condition does the result hold in the later case?

**Limitations:**

The authors discussed about the limitations that the theory requires diagonal weights and small initialization. However, it is not discussed whether dynamics (3) could cover optimizers in practical cases. I don't think this paper would have potential negative societal impact.

---

> ### Author Rebuttal · Authors · 2023-08-10
>
> Thank you for your feedback. We answer all your questions and address all your comments on weaknesses below. We hope that our responses will be sufficient to clear up any concerns and confusion. We would be happy to answer any more questions if you have them.
>
> * Q: On NLP transformers
>     * Thanks for the suggestion. We have **added experiments on NLP transformers** (GPT-2 trained on Wikitext) which show the exact same behavior; this is to be expected since the architecture is quite similar; see attached document.
> * Q: “How small is a ”small initialization“ in practice?”
>     * We observe gradual rank increase dynamics at the initialization scales used in standard practice (see our experiments on ViTs and on GPT-2). Furthermore, Figures 2 and 4 show that the gradual rank increase dynamics are even more pronounced if we take a smaller initialization scale, which is consistent with our theory.
>     * For an indication that we might observe these dynamics at practical initialization scales, see Figures 6, 7, and 8. These figures show that in our toy model there is already some stage-wise learning behavior at initialization scale 0.1 which is a quite practical scale. We will emphasize this in the revision by moving Figure 6 to the main text.
> * Q: “Dynamic (3) seems to hold only for gradient descent training, how does it apply to other optimizers like SGD/Adam?”
>     * For our theory, we analyze gradient flow training, which can be obtained as a limit of SGD or GD training with learning rate → 0 (see e.g., [Bach ’20]). Gradient flow is generally simpler to analyze than SGD, and it is a popular testbed for studying learning dynamics (see e.g., [Saxe et al. ‘13], [Arora et al. ’18], [Razin and Cohen ’20], to name just a few examples). Analysis of Adam or constant-step-size SGD is certainly an interesting question for future work. However, it is beyond the theoretical scope of this paper as (1) it would significantly increase the complexity of the analysis, which is already involved; (2) our analysis of gradient flow is already a significant novel contribution in view of the existing literature.
>     * Our experiments show that gradual rank increase dynamics hold with Adam training. For the revision we have **added experiments on SGD-trained transformers** which show the same behavior; see attached document.
> * Q: “The learning dynamics theories... the later case?”
>     * There are several works exploring how to transfer guarantees from gradient flow to GD (see e.g., [De Sa et al. '22]). In our case, the loss function is smooth, so by Gronwall’s inequality our main theorem holds automatically for GD with sufficiently small step size. We will add a remark in the main text.
> * Weakness: “The assumptions are very strong....”
>     * On diagonal weights: Diagonal linear networks have recently been used actively as a toy model from which useful insights can be drawn and for which rigorous results can be potentially derived. This line of work has been active for several years, primarily in NeurIPS,  ICML and COLT (see the various references in the paper), and remains an active area with several open problems on some of the simplest models. Our result in this line of work reaches a new level by obtaining a formal result for a transformer-inspired model that maintains the softmax; this is an important component as diagonal networks with a single non-linearity are still far from being well understood theoretically.
>     * On initialization scale: see our response to your questions above.
>     * On assumptions 4.2, 4.3, 4.4: These assumptions indeed hold for our toy model (an attention mechanism with diagonal weights). Please see Appendix C for experimental verification of these assumptions in our toy model.
> * Weakness: “The presentation of the paper could be improved.“
>     * Thank you for your feedback on the presentation of Sections 4.1.1 and 4.1.2. We were faced with a typical problem for theoretical papers: how to balance rigor and reader-friendliness in our proof sketch. There is a rigorous proof in Appendix A, so we decided to be less explicit on the error/magnitude bounds in the main text. We see now that this can be confusing for some readers. We will improve the presentation of Section 4 in the revision:
>         * As suggested by Reviewer yLKb, we will move Figure 6 to the main text. This illustrates Theorem 4.5 more clearly than Figure 3.
>         * As suggested by Reviewer yLKb, we will move some panels from Figures 9 and Figure 10 to the main text, since these illustrate assumptions 4.2,4.3,4.4 and our experimental verification.
>         * We will avoid phrases like “very small” and “negligible”, which referred to order $o_{\\alpha}(1)$ in Lines 150, 170, 188, 191, 196.
>         * We will improve the proof sketch in 4.1.1 and 4.1.2. This means adding more substance to the proof sketch in Section 4.1.1, and also shortening the proof sketch in Section 4.1.2 by removing technicalities. Concretely, in Section 4.1.1., we will explain how we use the conservation law to understand when the approximation to the dynamics is valid, since this is an important element.
> * References
>     * [Bach ’20] "Effortless optimization through gradient flows." Machine Learning Research Blog. https://francisbach.com/gradient-flows (2020).
>     * [Saxe et al. ’13] "Exact solutions to the nonlinear dynamics of learning in deep linear neural networks." arXiv preprint arXiv:1312.6120 (2013).
>     * [Arora et al. ’18] "On the optimization of deep networks: Implicit acceleration by overparameterization." International Conference on Machine Learning. PMLR, 2018.
>     * [Razin and Cohen ’20] "Implicit regularization in deep learning may not be explainable by norms." Advances in neural information processing systems 33 (2020): 21174-21187.
>     * [De Sa et al. ‘22] "From Gradient Flow on Population Loss to Learning with Stochastic Gradient Descent." Advances in Neural Information Processing Systems 35 (2022): 30963-30976.

---

> > ### Comment · Reviewer_ZvU1 · 2023-08-21
> >
> > Thanks for the detailed rebuttal. That does clear some of my confusions. I have increased my score accordingly. However, I still think the paper has a large room for improvement in its presentation. Overall I am still inclined for reject but won't be unset if it's accepted with major refactoring.

---

> > > ### Author Response · Authors · 2023-08-21
> > >
> > > Thank you for adjusting your score. In the revision, we will implement Reviewer yLKb's suggestions for improving the presentation. We will also implement the presentation changes (mostly in Section 4) that we promised in our response to you above. Apart from these, are there any other places where you would suggest changes to the presentation?

---

### Official Review · Reviewer_ryvs · 2023-07-26

**Soundness:** 3 good
**Presentation:** 3 good
**Contribution:** 3 good
**Rating:** 6
**Confidence:** 3

**Summary:**

The article "Transformers learn through gradual rank increase" considers the dynamics of training for the neural networks models with attention mechanism. The authors relate the training dynamics to a particular type of gradient flow. They show under 3 important assumptions:
i. diagonal weight matrices
ii. initialization is small
iii. only one coordinate is in "active" regime at a time
that dynamics occur in discrete stages:
(1) during most of each stage, the loss plateaus because the weights remain close to a saddle point
(2) at the end, the saddle point is quickly escaped and the rank of the weights increases by at most one

The developed theory is illustrated with a series of experiments.

**Strengths:**

1. Nice and clean results that show us the dynamics of neural networks with attention mechanism

2. Well connected with the previous research on training dynamics

3. Paper is well written

**Weaknesses:**

Assumptions are quite restrictive. It would be nice to relax the assumptions, especially assumption #3.

**Questions:**

1. Is the paper more about attention in general than specifically about transformers?

2. What will happen if some weights activate simultaneously? How the theory would be affected?

**Limitations:**

Limitations are adequately discussed

---

> ### Author Rebuttal · Authors · 2023-08-10
>
> Thank you for the positive review of our paper and your questions about the theory. We are glad that you found the results interesting and found our paper to be well-written. Please find below our response to your questions:
>
> * Q1: "Is the paper more about attention in general than specifically about transformers?"
>     * Our paper has two novel contributions: an experimental contribution and a theoretical contribution.
>         * The experimental contribution is that we observe that the difference between trained and initial weights grows in rank. The experiments are conducted on ViT and GPT2 transformers (which have attention layers inside). So the experiments are about transformers in practice.
>         * Our theory on the training dynamics applies generally to nonlinear networks with diagonal parameterization. Transformers with diagonal weights are the important special case on which we focus. Our theory applies to these transformers if the attention layers are the only trainable parameters (see Example 3.2 in our paper for details). So the theory applies to transformers of any depth, where only the attention layers are being trained.
> * Q2: “What will happen if some weights activate simultaneously? How the theory would be affected?“
>     * If more than one weight can activate simultaneously, then it becomes more burdensome to write down the dynamics in Algorithm 1. Nevertheless, we believe that we might be able to prove the multiple-weight-activation dynamics are valid if we also modify Assumption 4.4 appropriately. Thankfully, multiple weights do not seem to activate at exactly the same time in practice (see our experiments in Appendix C), so this non-degeneracy assumption seems to be a valid simplifying assumption.
> * Weakness: “Assumptions are quite restrictive. It would be nice to relax the assumptions, especially assumption #3.”
>     * We wholeheartedly agree that it would be nice to relax the assumptions 4.2, 4.3, and 4.4, but we also recall that they are validated on our toy model in Appendix C. Assumption 4.2 is mainly the assumption that two weights do not activate simultaneously, so see our answer above for how we believe this could be relaxed. We do not know how to relax Assumptions 4.3 and 4.4, but strong assumptions along these lines seem inevitable because of the generality of the theorem. In the revision, following the suggestion of reviewer yLKb, we will move some figures from Appendix C to the main text because these help illustrate the assumptions and our experimental verifications of them on the toy attention model.

---

> > ### Comment · Reviewer_ryvs · 2023-08-11
> > **Rebuttal answer**
> >
> > Thanks for answering my questions. I tend to keep my score as my thoughts on the paper didn't change after reading other reviews and your answers.

---

### Author Rebuttal · Authors · 2023-08-10

We thank the reviewers for their generally positive evaluations and for their helpful feedback that has helped us improve the paper in the revision. As suggested by the reviewers, we have **added new experiments**:
* on NLP transformers (GPT2 trained on Wikitext)
* and SGD-trained transformers (ViTs on CIFAR-10/100).

See attached file. These experiments further confirm the observations of the paper. We have responded to all reviewers’ questions individually, and we are happy to respond to any other questions they may have.

---

### Decision · Program_Chairs · 2023-09-21

**Decision:**

Accept (poster)

**Comment:**

The paper received mixed reviews initially (3666). After rebuttal and discussion, the negative reviewer raised the score to 4, citing mainly remaining doubts on presentation. Even though doubts remain about the implications in practical cases, where the strong assumptions are not met (ZvU1), the AC agrees that the paper provides new insights on the incremental learning dynamics in transformers. Therefore, given the positive evaluation of 3 reviewers, and the minor reservations of the negative reviewer, the AC recommends acceptance of the paper. The authors should incorporate the discussion in their final version.